EMBO
Molecular Medicine

# A novel thermoregulatory role for PDE10A in mouse and human adipocytes

Mohammed K Hankir[1], Mathias Kranz[2], Thorsten Gnad[3], Juliane Weiner[1], Sally Wagner[2], Winnie Deuther-Conrad[2], Felix Bronisch[1], Karen Steinhoff[4], Julia Luthardt[4], Nora Klöting[1], Swen Hesse[1,4], John P Seibyl[5], Osama Sabri[1,4], John T Heiker[1], Matthias Blüher[1], Alexander Pfeifer[3], Peter Brust[2] & Wiebke K Fenske[1,*]

## Abstract

Phosphodiesterase type 10A (PDE10A) is highly enriched in striatum and is under evaluation as a drug target for several psychiatric/neurodegenerative diseases. Preclinical studies implicate PDE10A in the regulation of energy homeostasis, but the mechanisms remain unclear. By utilizing small-animal PET/MRI and the novel radioligand [$^{18}$F]-AQ28A, we found marked levels of PDE10A in interscapular brown adipose tissue (BAT) of mice. Pharmacological inactivation of PDE10A with the highly selective inhibitor MP-10 recruited BAT and potentiated thermogenesis *in vivo*. In diet-induced obese mice, chronic administration of MP-10 caused weight loss associated with increased energy expenditure, browning of white adipose tissue, and improved insulin sensitivity. Analysis of human PET data further revealed marked levels of PDE10A in the supraclavicular region where brown/beige adipocytes are clustered in adults. Finally, the inhibition of PDE10A with MP-10 stimulated thermogenic gene expression in human brown adipocytes and induced browning of human white adipocytes. Collectively, our findings highlight a novel thermoregulatory role for PDE10A in mouse and human adipocytes and promote PDE10A inhibitors as promising candidates for the treatment of obesity and diabetes.

**Keywords** adipose tissue; energy expenditure; obesity; PDE10A; PET
**Subject Category** Metabolism

## Introduction

Obesity poses a tremendous burden for patients and healthcare systems alike for which safe and effective pharmacotherapies are still lacking. Due to the adverse side effects of most centrally acting appetite-suppressing compounds, alternative strategies that target energy-expending peripheral pathways have long been pursued by the pharmaceutical industry but with limited success (Cunningham & Wiviott, 2014).

The discovery of the persistence of thermogenic brown adipose tissue (BAT) in adult humans has rekindled interest in pharmacologically increasing energy expenditure as a means to promote weight loss (Sidossis & Kajimura, 2015). This is further enthused by the increasing number of factors identified, which induce "browning" of energy-storing white adipose tissue (WAT) to energy-consuming beige adipose tissue (BeAT) (Seale, 2015). A defining attribute of brown/beige adipocytes is their expression of uncoupling protein 1 (UCP-1) (Wu *et al*, 2012), which disrupts the proton gradient across the inner mitochondrial membrane, thereby funneling energy to produce heat instead of ATP (Shabalina *et al*, 2013). The identification of new regulatory pathways of thermogenesis forms an important basis for the development of novel BAT- and/or BeAT-centered therapies.

Cyclic nucleotides are ubiquitous second messengers involved in both short-term and long-term intracellular signaling pathways. In adipocytes, cyclic adenosine monophosphate (cAMP) and cyclic guanosine monophosphate (cGMP) stimulate lipolysis, through protein kinase A (PKA) and protein kinase G (PKG), respectively (Cannon & Nedergaard, 2004; Lafontan *et al*, 2008), which in brown/beige adipocytes rapidly activates UCP-1 (Li *et al*, 2014). The UCP-1 enhancer and promoter harbor various domains that bind to transcription factors recruited to the nucleus downstream of PKA/PKG signaling, and thus, elevations of cAMP and cGMP in the cytosol also induce UCP-1 transcription (Haas *et al*, 2009; Nikolic *et al*, 2011; Bordicchia *et al*, 2012; Mitschke *et al*, 2013; Hoffmann *et al*, 2015).

Phosphodiesterases (PDEs) catalyze the breakdown of cAMP and cGMP serving to fine-tune their function. In total, 21 isozymes encoded by 11 different genes have been characterized, which share similar overall structural organization with conserved catalytic and

1  Integrated Research and Treatment Centre for Adiposity Diseases, University Hospital, University of Leipzig, Leipzig, Germany
2  Institute of Radiopharmaceutical Cancer Research, Helmholtz-Zentrum Dresden-Rossendorf, Neuroradiopharmaceuticals, Leipzig, Germany
3  Institute of Pharmacology and Toxicology, University Hospital, University of Bonn, Bonn, Germany
4  Department of Nuclear Medicine, University Hospital, University of Leipzig, Leipzig, Germany
5  Molecular NeuroImaging, LLC, New Haven, CT, USA
  *Corresponding author. Tel: +49 341 97 13306; Fax: +49 341 97 13369; E-mail: wiebkekristin.fenske@medizin.uni-leipzig.de

regulatory domains, but differ in their modes of inhibition and stimulation, substrate specificity, cellular and tissue distribution profiles (Keravis & Lugnier, 2010). Various PDEs are expressed in both brown and white adipose tissue depots (Omar *et al*, 2011; Kraynik *et al*, 2013). In WAT, PDE3B has received the most attention as a downstream target of insulin's anti-lipolytic effects through Akt-mediated phosphorylation and activation (Armani *et al*, 2011). Preclinical studies have also shown that long-term inhibition of PDEs 1, 4, and 5 promotes a negative energy balance by increasing energy expenditure through either increased BAT function and/or browning of WAT (Ayala *et al*, 2007; Park *et al*, 2012; Mitschke *et al*, 2013; Pan *et al*, 2014). Indeed, the weight lowering effect of resveratrol has been attributed in part to its ability to inhibit PDEs in WAT, thereby increasing cAMP levels (Park *et al*, 2012). However, these PDEs are widely distributed and so off-target effects of selective enzyme inhibitors may preclude their utility as obesity treatments.

Of the PDEs, the dual-specificity phosphodiesterase PDE10A has the most restricted tissue distribution being mainly expressed in the striatum (Seeger *et al*, 2003; Coskran *et al*, 2006; Xie *et al*, 2006; Lakics *et al*, 2010; Jager *et al*, 2012), a brain region involved in motor, emotional, and cognitive functions (Macpherson *et al*, 2014; Floresco, 2015). Selective inhibitors of PDE10A were initially considered for the treatment of schizophrenia due to their efficacy in preclinical models and are now being evaluated for a number of psychiatric and movement disorders (Wilson & Brandon, 2015). An additional role for PDE10A in the regulation of energy balance has been implicated in PDE10A KO mice (Siuciak *et al*, 2006; Nawrocki *et al*, 2014). These mice display reduced food intake specifically to a high–fat, high-carbohydrate (HFHC) diet and are resistant to diet-induced obesity (DIO) (Nawrocki *et al*, 2014). Furthermore, chronic administration of the selective PDE10A inhibitor THPP-6 to DIO mice causes weight loss beyond that which occurs with pair-fed animals and prevents the concurrent drop in oxygen consumption (Nawrocki *et al*, 2014). Together, these findings suggest that chronic PDE10A inhibition causes weight loss by both reducing hedonic feeding and increasing energy expenditure; however, the underlying mechanisms are poorly defined.

Here, we describe a novel thermoregulatory role for PDE10A in mouse and human adipocytes. By implementing small-animal PET-MRI, we discovered marked and restricted uptake of the specific PDE10A radioligand [18F]-AQ28A (Wagner *et al*, 2016) by interscapular BAT in mice, indicating for the first time expression of PDE10A in this thermogenic tissue. Consistent with this finding, we found that acute pharmacological suppression of PDE10A enzymatic activity with the highly selective inhibitor MP-10, which has over 1,000-fold specificity for PDE10A over other PDEs (Grauer *et al*, 2009), sufficiently elicits multiple functional responses in BAT *in vivo*. In addition, we found that acute inhibition of PDE10A with MP-10 potently causes biochemical effects in visceral white adipose tissue *ex vivo*. A cell-autonomous role for PDE10A in murine brown adipocytes was further demonstrated in acute *in vitro* cell culture experiments. In chronic studies, the inhibition of PDE10A with MP-10 caused weight loss in diet-induced obese mice associated with increased energy expenditure and browning of visceral white adipose tissue. Providing a translational dimension to the current work, PET imaging further revealed high levels of PDE10A in the supraclavicular BAT and abdominal SAT regions in humans. Finally, cell culture experiments revealed that the inhibition of PDE10A with MP-10 stimulates thermogenic gene expression in primary human brown adipocytes and causes browning of primary human white adipocytes.

# Results

## PDE10A is expressed in brown adipose tissue of mice

We performed small-animal PET/MRI on normal weight mice that received the newly developed PDE10A radioligand [18F]-AQ28A (Wagner *et al*, 2016). Fused PET/MRI images revealed intense symmetrical uptake of [18F]-AQ28A in interscapular BAT with minimal uptake in surrounding skeletal muscle (Fig 1). Pronounced radionuclide accumulation was also evident in liver (Fig 1) likely representing metabolites of [18F]-AQ28A (Fig 1). Dynamic PET data analysis confirmed higher PDE10A levels in BAT compared to skeletal muscle ($P < 0.05$ from the 20th minute post-injection, Fig 1). As a PDE10A-specific radioligand which crosses the blood–brain barrier, [18F]-AQ28A also exhibited marked and restricted bilateral accumulation in striatum following systemic administration with minimal uptake in hypothalamus (Appendix Fig S1). We additionally performed a 'blocking' experiment in which mice received the highly selective PDE10A inhibitor MP-10 (Grauer *et al*, 2009) prior to receiving [18F]-AQ28A. PET images revealed substantially reduced [18F]-AQ28A in BAT as a result of competition with MP-10 (Appendix Fig S2). Together, these data indicate that PDE10A is markedly expressed in BAT in addition to striatum in mice.

## PDE10A regulates brown and visceral white adipose tissue thermogenic function in mice

To test the functional significance of PDE10A expression in thermogenic BAT, pharmacological [18F]-FDG-PET experiments with MP-10 were performed on normal weight and diet-induced obese (DIO) mice. Acute administration of MP-10 (30 mg/kg) resulted in significantly higher [18F]-FDG uptake by BAT compared to vehicle treatment under both fasted ($P < 0.01$; Fig 2A) and *ad libitum* fed conditions ($P < 0.05$; Appendix Fig S3A) in normal weight but not DIO mice (Appendix Fig S3B). To ascertain whether increased BAT metabolic function in response to PDE10A inhibition is associated with enhanced temperature-induced thermogenesis, core body temperature was measured during a 4-h cold challenge. Acute administration of MP-10 (30 mg/kg) protected against hypothermia compared to vehicle treatment ($P < 0.05$ for all time-points; Fig 2B). Next, to determine a cell-autonomous function of PDE10A, cell culture experiments were performed. Acute treatment of differentiated brown adipocytes derived from murine interscapular BAT with 100 nM MP-10 markedly stimulated lipolysis ($F_{(6, 21)} = 70.08$, $P < 0.0001$; 1.58-fold higher glycerol release than control, $P < 0.0001$; Appendix Fig S4A). The amount of glycerol released from cells in response to 100 nM MP-10 was significantly higher than for 10 μM cAMP ($P < 0.05$) and 10 μM cGMP ($P < 0.05$) treatments, but significantly lower than for control treatment with 1 μM norepinephrine (NE) ($P < 0.0001$; Appendix Fig S4A). Notably, the stimulation of lipolysis by MP-10 (100 nM) was similar to that of dual-cyclic nucleotide treatment ($P = 0.99$; Appendix Fig S4A).

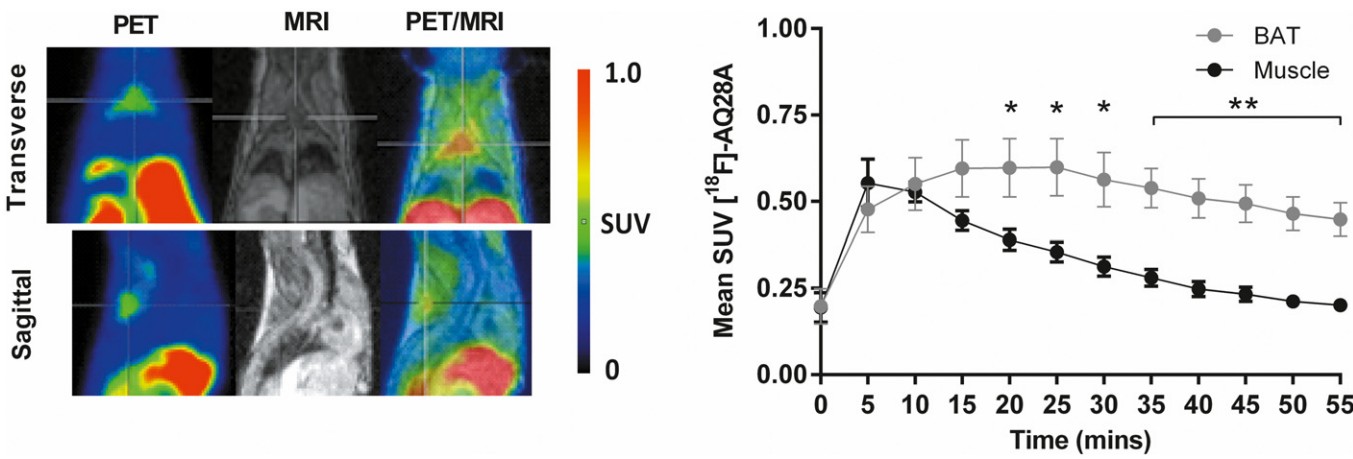

**Figure 1. PDE10A is expressed in brown adipose tissue of mice.**

Representative PET, MRI, and fused PET/MRI images of the thoracic region of a lean mouse that received the PDE10A radioligand [$^{18}$F]-AQ28A. Interscapular brown adipose tissue (BAT) in each image is highlighted by the crosshairs. The mean standardized uptake (SUV) value of [$^{18}$F]-AQ28A in BAT and skeletal muscle throughout the dynamic scan was calculated and compared for each animal ($n = 5$). *$P = 0.0488$ (20 min), *$P = 0.0236$ (25 min), *$P = 0.0167$ (30 min), **$P = 0.00301$ (35 min), **$P = 0.00269$ (40 min), **$P = 0.00215$ (45 min), **$P = 0.00129$ (50 min), and **$P = 0.00141$ (45 min) using unpaired 2-tailed Student's $t$-tests. Data are represented as mean ± SEM.

We then compared relative mRNA expression of *Pde10a* in various adipose tissue depots. Real-time quantitative PCR analysis (RT–qPCR) revealed a 4.7-fold ($P < 0.01$) and 17.6-fold ($P < 0.01$) higher level of expression of *Pde10a* in peri-ovarian VAT and interscapular BAT compared to inguinal SAT, respectively (Fig 2C). Expression of *Pde10a* mRNA was also significantly higher in BAT compared to VAT ($P < 0.05$; Fig 2C). To address the specificity of the *Pde10a* mRNA expression profile in adipose tissue, *Pde3b* was additionally analyzed. Expression of *Pde3b* mRNA was similar in peri-ovarian VAT and 3.8-fold ($P < 0.01$) higher in interscapular BAT compared to inguinal SAT, respectively (Appendix Fig S5B). Expression of *Pde3b* mRNA was also significantly higher in BAT compared to VAT ($P < 0.01$; Appendix Fig S5B). In brain, a 6.8-fold ($P < 0.001$; Appendix Fig S6A) higher expression of *Pde10a* in striatum compared to hypothalamus was found, whereas *Pde3b* levels were approximately equal in the two regions (Appendix Fig S5A). Corresponding to *Pde10a* expression levels in adipose tissue depots, acute administration of MP-10 (30 mg/kg) increased cyclic nucleotide concentrations in interscapular BAT and peri-ovarian VAT but not inguinal SAT (Fig 2D). This was accompanied with increased mRNA expression of the thermoregulatory genes peroxisome proliferator-activated receptor gamma coactivator-1 alpha (*Pgc1alpha*), *Ucp1*, cell death activator (*Cidea*), and PR domain containing 16 (*Prdm16*) in BAT and VAT, but not in SAT (Fig 2E). Notably, the inhibition of PDE10A with MP-10 also reduced *Pde10a* mRNA expression in VAT and BAT (Appendix Fig S7B) with no effect on that of *Pde3b* (Appendix Fig S7A). In brain, acute administration of MP-10 (30 mg/kg) induced mRNA expression of the nuclear protein *Fos*, the transcription factor zinc finger 268 (*Zif268*), and the neuropeptide precursor preproenkephalin (*Ppe*) in striatum (Appendix Fig S6C), whereas mRNA expression of *Fos*, the feeding neuropeptides agouti-related peptide (*Agrp*), neuropeptide Y (*Npy*), and proopiomelanocortin (*Pomc*) in hypothalamus were unaffected (Appendix Fig S6B). Notably, mRNA expression of the neuropeptide precursor preprodynorphin (*Ppd*) was not induced in striatum in

response to acute inhibition of PDE10A (Appendix Fig S6C). As with VAT and BAT, the inhibition of PDE10A with MP-10 reduced *Pde10a* mRNA expression in striatum (Appendix Fig S7B) with no effect on *Pde3b* mRNA expression (Appendix Fig S7A). Together, these data suggest that PDE10A regulates interscapular BAT and peri-ovarian VAT thermogenic potential independent of hypothalamic function.

## Expression of PDE10A is increased in brown adipose tissue in different mouse models of obesity

To evaluate whether expression of PDE10A changes as a function of body weight, [$^{18}$F]-AQ28A uptake in BAT and striatum was measured on high–fat, high-sugar (HFHS) DIO and genetically obese (leptin deficient) mouse models. At the time of scans, normal weight mice weighed $26.8 \pm 1.4$ g, DIO mice weighed $41.1 \pm 1.1$ g and ob/ob mice weighed $55 \pm 1.1$ g ($P < 0.05$ for lean vs. DIO and $P < 0.001$ for lean vs. ob/ob using unpaired Student's $t$-tests). PET data analysis revealed higher PDE10A levels in interscapular BAT *in vivo* of DIO and ob/ob mice compared to normal weight mice ($P < 0.05$ for lean vs. DIO and $P < 0.01$ for lean vs. ob/ob; Fig 3). In brain, PET data analysis additionally revealed higher PDE10A levels in striatum of ob/ob mice compared to normal weight mice ($P = 0.0001$; Appendix Fig S8). Furthermore, PDE10A levels in striatum of ob/ob mice were higher than in DIO mice ($P < 0.01$; Appendix Fig S8). Together, these data suggest that PDE10A expression increases in interscapular BAT and striatum in various models of obesity in mice.

## Chronic inhibition of PDE10A with MP-10 causes weight loss in DIO mice and improves insulin sensitivity associated with browning of visceral white adipose tissue

Having demonstrated the presence and functionality of PDE10A in interscapular BAT and peri-ovarian VAT in regulating adaptive

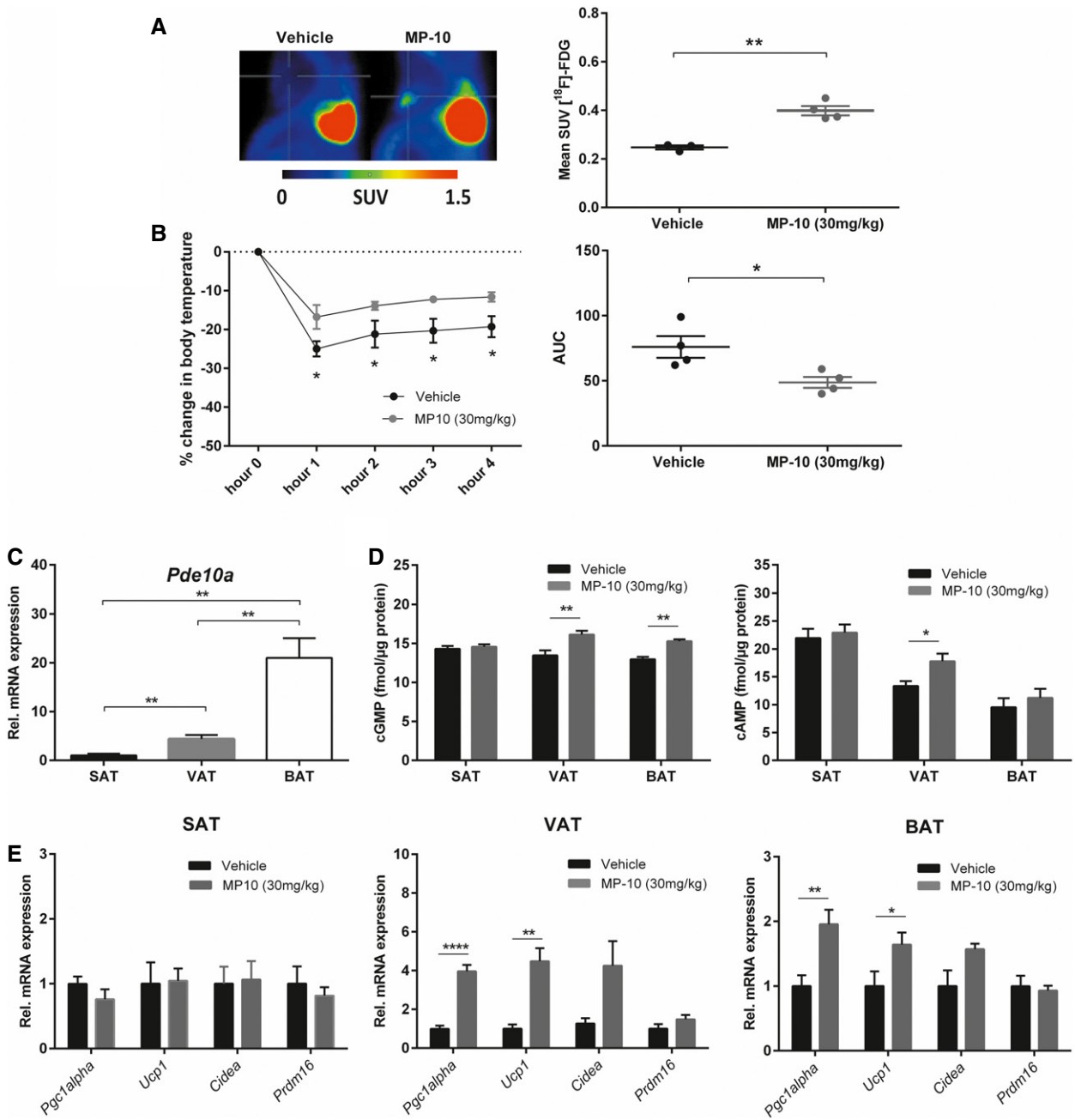

**Figure 2.  Acute pharmacological inhibition of PDE10A with MP-10 stimulates *in vivo* glucose uptake by brown adipose tissue and potentiates thermogenesis in lean mice.**

A   Representative PET images of a sagittal view of the thoracic region of fasted lean mice treated with either MP-10 (30 mg/kg) (*n* = 3) or vehicle (*n* = 4) prior to receiving [$^{18}$F]-FDG. Interscapular brown adipose tissue (BAT) in each image is highlighted by the crosshairs. The mean standardized uptake value (SUV) of [$^{18}$F]-FDG in BAT after both treatments was calculated for each group. **$P$ = 0.0013 using unpaired 2-tailed Student's *t*-test.

B   The percentage (%) change in core body temperature and associated area under the curve (AUC) of lean mice treated with MP-10 (30 mg/kg) (*n* = 4) or vehicle (*n* = 4) prior to a cold challenge study for 4 h at 8°C. *$P$ = 0.0108 (1$^{st}$ hour), *$P$ = 0.0218 (2$^{nd}$ hour), *$P$ = 0.0121 (3$^{rd}$ hour), *$P$ = 0.0168 (4$^{th}$ hour) and *$P$ = 0.0264 (AUC) using unpaired 2-tailed Student's *t*-tests.

C   RT–qPCR analysis of relative mRNA expression of *Pde10a* in BAT, peri-ovarian visceral white adipose tissue (VAT), and inguinal subcutaneous white adipose tissue (SAT) of lean mice (*n* = 5). **$P$ = 0.0063 (SAT vs. VAT), **$P$ = 0.0011 (SAT vs. BAT) and **$P$ = 0.0037 (VAT vs. BAT) using unpaired 2-tailed Student's *t*-test.

D   Cyclic GMP (cGMP) and cyclic AMP (cGMP) concentrations normalized to protein concentration in SAT, VAT, and BAT 30 min after treatment of lean mice with MP-10 (30 mg/kg) (*n* = 6) or vehicle control (*n* = 6). **$P$ = 0.0085 (cGMP in VAT), **$P$ = 0.0011 (cGMP in BAT), and *$P$ = 0.04 (cAMP in VAT) using unpaired 2-tailed Student's *t*-tests.

E   RT–qPCR analysis of relative mRNA expression of thermogenic genes in SAT, VAT, and BAT of lean mice treated with MP-10 (30 mg/kg) (*n* = 7) or vehicle control (*n* = 6). ****$P$ < 0.0001 (*Pgc1alpha* in VAT), **$P$ = 0.0012 (*Ucp1* in VAT), **$P$ = 0.0071 (*Pgc1alpha* in BAT), and *$P$ = 0.0486 (*Ucp1* in BAT) using unpaired 2-tailed Student's *t*-tests.

Data information: Data are represented as mean ± SEM. Graphs in (A, right panel) and (B, right panel) show individual data where bars represent mean ± SEM.

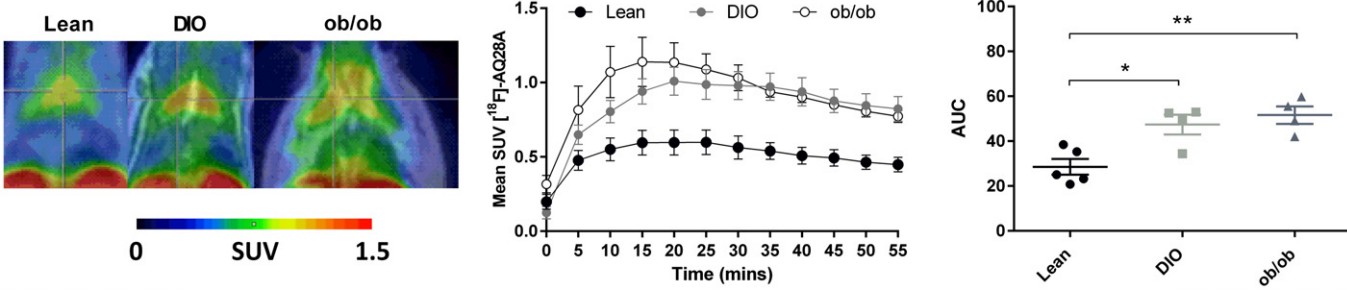

**Figure 3. Expression of PDE10A in brown adipose tissue is increased in different mouse models of obesity.**
Representative fused PET/MRI images of the transverse view of the thoracic region of lean ($n = 5$), diet-induced obese (DIO) ($n = 4$), and leptin-deficient (ob/ob) ($n = 4$) mice that received the PDE10A radioligand [$^{18}$F]-AQ28A. Interscapular brown adipose tissue (BAT) in each image is highlighted by the crosshairs. The mean standardized uptake (SUV) value of [$^{18}$F]-AQ28A in BAT throughout the dynamic scan was calculated with associated area under the curve (AUC). *$P = 0.0128$ (lean vs. DIO) and **$P = 0.0031$ (lean vs. ob/ob) using unpaired 2-tailed Student's $t$-test. Data are represented as mean $\pm$ SEM. Graph in right panel shows individual data where bars represent mean $\pm$ SEM.

thermogenesis, its possible involvement in body weight regulation and glucose metabolism was then assessed in metabolic studies performed on two strains of DIO mice. Once-daily administration of MP-10 (10 mg/kg) to HFHS DIO CD-1 mice resulted in significant weight loss compared to vehicle-treated controls from day 4 of treatment until conclusion of the study at day 7 (main effect of treatment $F_{(1, 48)} = 56.07$, $P < 0.0001$; main effect of time $F_{(7, 48)} = 36.09$, $P < 0.0001$ and interaction $F_{(7, 48)} = 7.86$, $P < 0.0001$; Fig 4A). Mice treated with MP-10 (10 mg/kg) lost on average $8.61 \pm 0.37\%$ body weight vs. $3.31 \pm 0.68\%$ for controls ($P < 0.0001$; Fig 4A). Importantly, no difference in cumulative food intake between MP-10-treated and vehicle-treated groups was observed with this dose ($15.62 \pm 0.73$ g vs. $17.73 \pm 0.97$ g, respectively, $P = 0.19$; Fig 4B). Similarly, once-daily administration of MP-10 (10 mg/kg) to HF DIO C57BL/6 mice resulted in significant weight loss compared to vehicle-treated controls on day 6 and day 7 of treatment (main effect of treatment $F_{(1, 48)} = 19.47$, $P < 0.0001$; main effect of time $F_{(7, 48)} = 33.27$, $P < 0.0001$ and interaction $F_{(7, 48)} = 2.05$, $P = 0.06$; Appendix Fig S9A). Again, no difference in cumulative food intake between MP-10-treated and vehicle-treated groups was evident ($P = 0.86$; Appendix Fig S9B). An insulin tolerance test revealed that mice treated with MP-10 (10 mg/kg) compared to those

treated with vehicle control had significantly improved glucose handling ($P < 0.05$ from the 45th minute post-injection; Fig 4C). Indirect calorimetry showed that mice treated with MP-10 (30 mg/kg) compared to those treated with vehicle control had a higher respiratory quotient (RQ), indicative of increased carbohydrate metabolism ($P < 0.05$ between 2 am and 4 am; Fig 4D), as well as increased resting energy expenditure ($P < 0.01$ between midnight and 4 am; Fig 4E). Importantly, there was no difference in locomotor activity between MP-10-treated and vehicle-treated groups (Fig 4F).

When analyzing thermoregulatory gene induction in adipose tissue depots of mice following chronic MP-10 treatment, *Pgc1alpha*, *Ucp1*, and *Cidea* mRNA were upregulated in peri-ovarian VAT but not inguinal SAT (Fig 4G). These data are indicative of the adoption of a BeAT phenotype specifically for peri-ovarian VAT. In keeping with this, when assessing mRNA expression of the beige adipocyte-specific markers transmembrane protein 26 (*Tmem26*) and T box transcription factor 1 (*Tbx1*) (Wu *et al*, 2012), higher levels of *Tmem26* were found in peri-ovarian VAT ($P < 0.05$), but not in inguinal SAT ($P = 0.77$) of MP-10-treated animals (Fig 4G). We also performed immunohistochemical analysis of UCP-1 protein expression in inguinal SAT, peri-ovarian VAT, and interscapular

**Figure 4. Weight loss from chronic pharmacological inhibition of PDE10A with MP-10 is associated with browning of visceral white adipose tissue and improved insulin sensitivity in diet-induced obese mice.**

A, B    The percentage (%) change in body weight and cumulative food intake of diet-induced obese (DIO) mice in response to daily treatment with MP-10 (10 mg/kg) or vehicle for a week ($n = 4$ per group). *$P = 0.0264$ (day 4), ***$P = 0.0002$ (day 5), ***$P < 0.0001$ (day 6), and ***$P < 0.0001$ (day 7) using two-way analysis of variance (ANOVA) with Sidak's *post hoc* test.

C    Upon completion of the chronic feeding study, an insulin tolerance test was performed on DIO mice with single treatment of 0.75 U/kg insulin ($n = 4$ per group). *$P = 0.0246$ and *$P = 0.0305$ using unpaired 2-tailed Student's $t$-tests.

D–F    Upon completion of the chronic feeding study in a separate group of DIO mice (see Materials and Methods section), resting energy expenditure (REE), respiratory quotient (RQ), and locomotor activity of DIO mice treated with MP-10 (30 mg/kg) compared to vehicle control were measured in metabolic cage experiments ($n = 4$ per group). *$P = 0.0239$ (02:00, RQ), *$P = 0.0424$ (02:30, RQ), *$P = 0.0420$ (03:00, RQ), *$P = 0.0230$ (03:30, RQ), *$P = 0.0213$ (04:00, RQ), **$P = 0.0116$ (00:00, REE), **$P = 0.00823$ (00:30, REE), **$P = 0.00969$ (01:00, REE), **$P = 0.00274$ (01:30, REE), **$P = 0.00323$ (02:00, REE), **$P = 0.000441$ (02:30, RE), **$P = 0.00390$ (03:00, REE), **$P = 0.000211$ (03:30, REE), **$P = 0.00216$ (04:00, REE), and **$P = 0.0185$ (04:30, REE) using unpaired 2-tailed Student's $t$-tests.

G    RT–qPCR analysis of relative mRNA expression of candidate genes in inguinal subcutaneous white adipose tissue (SAT), peri-ovarian visceral white adipose tissue (VAT), and interscapular brown adipose tissue (BAT) ($n = 8$ per group). ***$P = 0.0004$ (*Pgc1alpha* VAT), **$P = 0.0036$ (*Ucp1* VAT), **$P = 0.0068$ (*Cidea* VAT), *$P = 0.0411$ (*Tmem26* VAT), and **$P = 0.0070$ (*Pgc1alpha* BAT) using unpaired 2-tailed Student's $t$-tests.

H    Representative sections of inguinal SAT, peri-ovarian VAT, and interscapular BAT immunostained for UCP1 of mice chronically treated with MP-10 (10 mg/kg) or vehicle control. Scale bars, 100 μm.

Data information: Data are represented as mean $\pm$ SEM. Graph in (B) shows individual data where bars represent mean $\pm$ SEM.

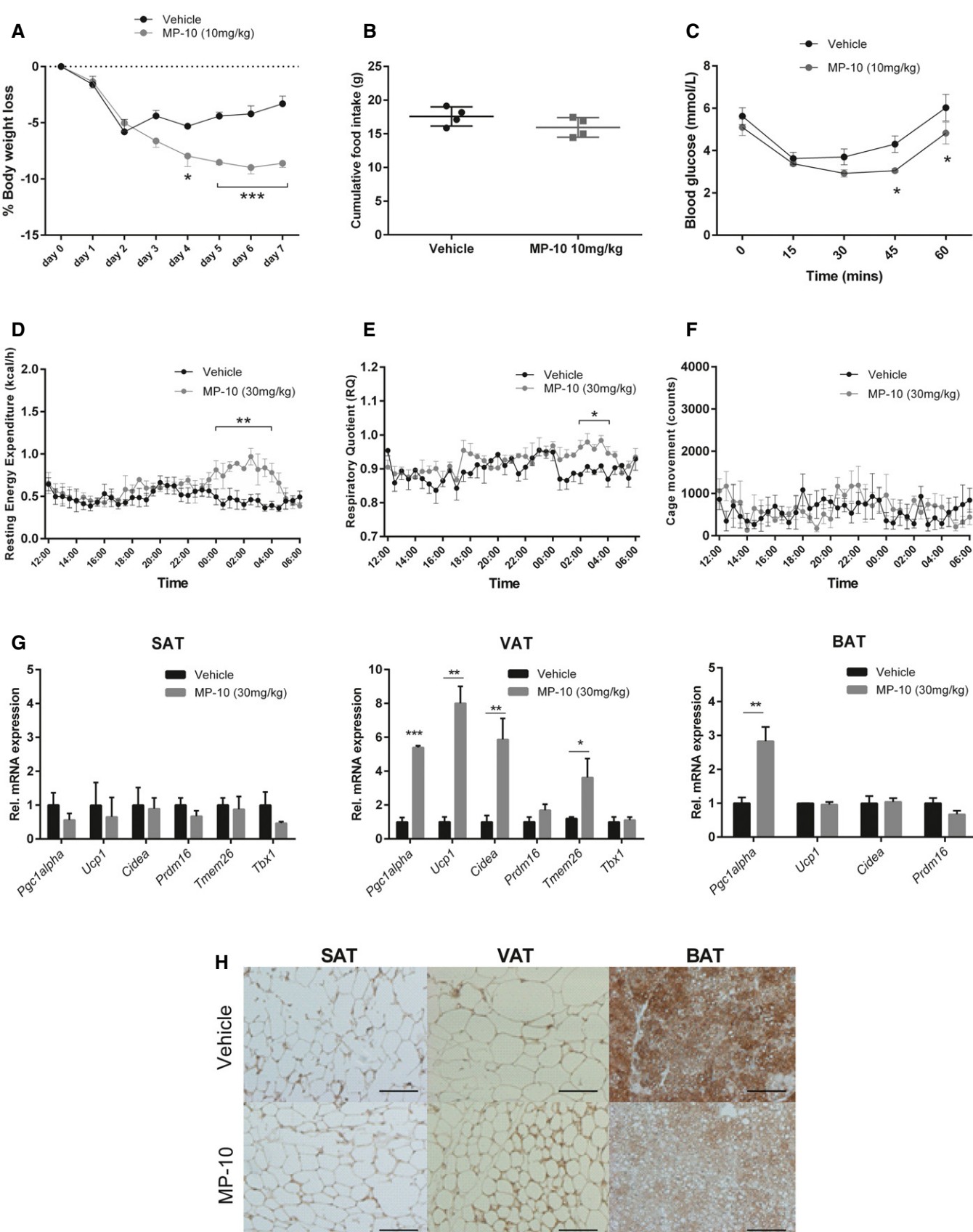

**Figure 4.**

BAT after chronic inhibition of PDE10A with MP-10 in DIO mice. Corresponding to the *mRNA* expression just described, increased immunostaining for UCP1 was found in VAT but not in SAT (Fig 4H). Adipocytes were also visibly smaller in VAT after chronic MP-10 treatment (Fig 4H). Interestingly, there was markedly decreased immunostaining for UCP-1 in BAT after chronic MP-10 treatment compared to vehicle control (Fig 4H). Together, these results suggest that weight loss in response to chronic PDE10A inhibition with a sub-anorectic dose of MP-10 can occur through increased energy expenditure mediated in part by browning of peri-ovarian VAT.

### Evidence for PDE10A expression in human brown and white adipose tissue *in vivo*

To translate our findings from animal studies, the expression of PDE10A in human BAT *in vivo* was evaluated. For this purpose, whole-body PET images obtained from previous dosimetry studies on subjects who received the specific PDE10A radioligand [18F]-MNI-659 were re-analyzed (Barret *et al*, 2014). PET data analysis revealed marked uptake of [18F]-MNI-659 in the supraclavicular region, but minimal uptake in skeletal muscle ($P < 0.001$ from the 5th minute post-injection to the 75th; Fig 5). Interestingly, analysis of each individual subject revealed higher [18F]-MNI-659 uptake in the supraclavicular region in individuals with a higher BMI (Appendix Fig S10). There was pronounced radionuclide accumulation in the abdominal cavity likely representing metabolites of [18F]-MNI-659 from the biliary route (Barret *et al*, 2014) (Appendix Fig S11). Marked accumulation of [18F]-MNI-659 in subcutaneous

abdominal SAT region could also be found (Appendix Fig S11). This retrospective analysis suggests for the first time that PDE10A is expressed in human supraclavicular BAT in addition to WAT depots.

### Inhibition of PDE10A with MP-10 stimulates thermogenic gene expression in human brown adipocytes and causes browning of human white adipocytes

Having established that PDE10A is expressed in human supraclavicular BAT and WAT depots, we next sought to determine its functional significance. We first evaluated mRNA expression of *Pde3b* and *Pde10a* in various undifferentiated and differentiated human cell types. When comparing between primary brown adipocyte, myocyte, and white adipocyte precursors, *Pde3b* mRNA expression predominated in white preadipocytes (Appendix Fig S12A), while *Pde10a* mRNA expression predominated in brown preadipocytes (Appendix Fig S12B). Interestingly, in mature cells mRNA expression of *Pde3b* completely disappeared with the exception of brown adipocytes (Appendix Fig S12A), while *Pde10a* expression significantly decreased ($P < 0.01$ in white adipocytes, $P < 0.01$ in myocytes and $P < 0.001$ in brown adipocytes) but remained detectable (Appendix Fig S12B). In line with the higher expression of PDE10A in brown preadipocytes compared to white, treatment of the former during differentiation with MP-10 (100 nM) increased mRNA expression of markers of adipogenesis peroxisome proliferator-associated receptor gamma (*Ppargamma*) ($P < 0.01$) and adipocyte protein 2 (*Ap2*) ($P < 0.01$) but not in the latter (Appendix Fig S12C and D).

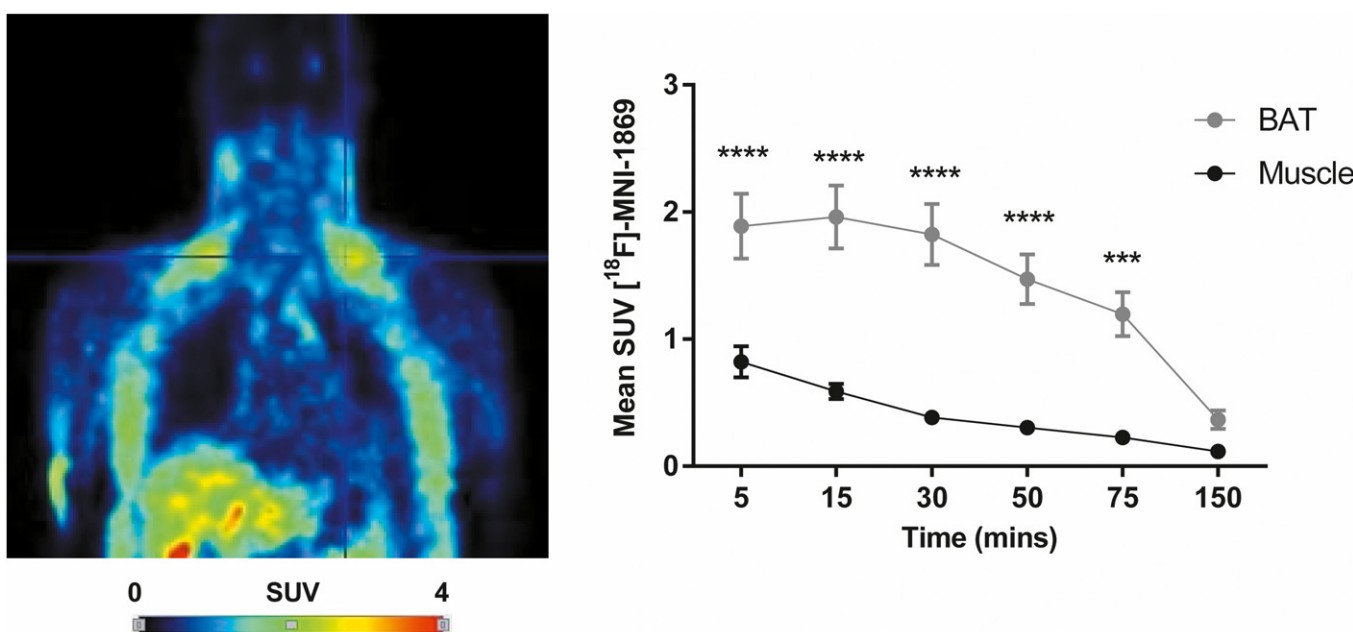

**Figure 5. PDE10A is expressed in supraclavicular brown tissue of humans.**
Representative PET image of the upper body of a human subject who received the PDE10A radioligand [18F]-MNI-1869. Supraclavicular brown adipose tissue (BAT) is highlighted by the crosshairs. The mean standardized uptake value (SUV) of [18F]-MNI-1869 in supraclavicular BAT and skeletal muscle was calculated ($n = 4$). **$P = 0.00210$ (5 min), ****$P < 0.0001$ (15 min), ****$P < 0.0001$ (30 min), ****$P < 0.0001$ (50 min), and **$P = 0.00462$ (75 min) using unpaired 2-tailed Student's *t*-tests. Data are represented as mean ± SEM.

In differentiated human primary brown adipocytes, acute treatment with MP-10 (100 nM) stimulated lipolysis ($F_{(6, 28)}$ = 28.99, $P < 0.0001$; 1.48-fold higher glycerol release than control, $P < 0.0001$; Appendix Fig S4B). As with murine brown adipocytes, the amount of glycerol released from cells in response to 100 nM MP-10 was significantly higher than for 10 μM cAMP ($P < 0.01$) and 10 μM cGMP ($P < 0.01$) treatments, but significantly lower than with 1 μM NE control treatment ($P < 0.001$; Appendix Fig S4B). Again as with murine brown adipocytes, the stimulation of lipolysis by MP-10 (100 nM) was similar to that of dual-cyclic nucleotide treatment ($P = 0.54$; Appendix Fig S4B).

Finally, to assess the consequence of long-term inhibition of PDE10A, analysis of mRNA expression of thermogenic genes in differentiated primary human brown and white adipocytes was then performed following chronic treatment with MP-10. In brown adipocytes, expression of *Ucp1* and type 2 deiodinase (*Dio2*) were most sensitively regulated by MP-10 with both doses (1 nM and 100 nM) markedly increasing expression ($F_{(5, 18)}$ = 19.90, $P < 0.0001$; 1.49-fold and 2.06-fold, respectively, for *Ucp1*, $P < 0.01$ and $P < 0.0001$;

1.90-fold and 2.34-fold, respectively, for *Dio2*, $P < 0.001$ and $P < 0.0001$; Fig 6A). High-dose MP-10 (100 nM) also increased expression of *Pgc1alpha* by 1.36-fold ($P < 0.05$) and *Cidea* by 1.64-fold ($P < 0.05$; Fig 6A). Notably, as is the case with acute systemic treatment of MP-10 to mice, applying high-dose MP-10 (100 nM) to human primary brown adipocytes also decreased *Pde10a* ($P < 0.01$; Appendix Fig S7D) but not *Pde3b* (Appendix Fig S7C) mRNA expression in these cells compared to control treatment. In human primary white adipocytes, expression of *Ucp1* and type 2 deiodinase (*Dio2*) were again the most sensitively regulated by MP-10 with both doses (1 and 100 nM) significantly increasing expression ($F_{(5, 18)}$ = 17.15, $P < 0.0001$; 1.34-fold and 1.79-fold, respectively, for *Ucp1*, $P < 0.05$ and $P < 0.0001$; 1.82-fold and 2.56-fold, respectively, for *Dio2*, $P = 0.32$ and $P = 0.01$; Fig 6B). Together, these results indicate a cell-autonomous role of PDE10A in regulating differentiation, lipolysis, and thermogenic gene expression in human adipocytes. Moreover, pharmacological inhibition of PDE10A with MP-10 stimulates thermogenic gene expression in human brown adipocytes and causes the browning of human white adipocytes.

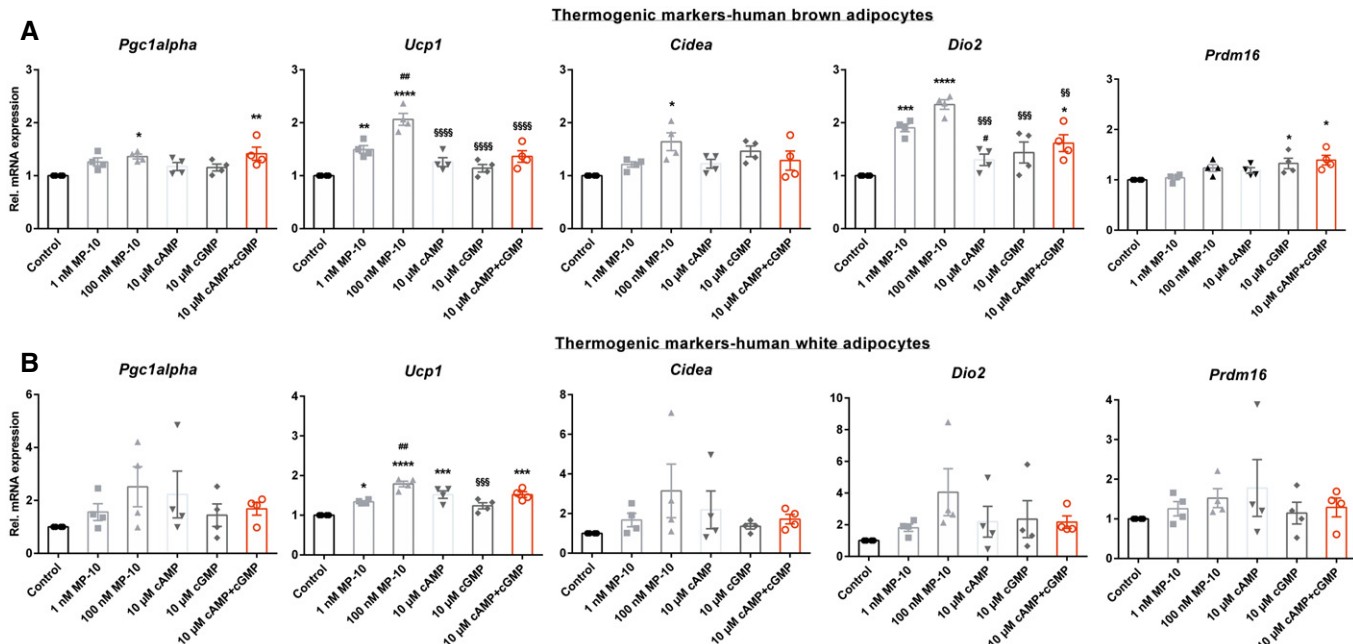

**Figure 6. Pharmacological inhibition of PDE10A with MP-10 stimulates thermogenic gene expression in human brown adipocytes and causes browning of human white adipocytes.**

A  RT–qPCR analysis of relative mRNA expression of thermogenic genes in cultured primary human brown adipocytes in response to chronic treatments (8 h) with DMSO (control), MP-10 (1 and 100 nM), cyclic AMP (cAMP—10 μM), cyclic GMP (cGMP—10 μM), and cAMP + cGMP (cA + cG—10 μM (n = 4 separate cultures). *P = 0.0278 vs. (100 nM MP-10 vs. control; *Pgc1alpha*), **P = 0.0097 (10 μM cA + cG vs. control; *Pgc1alpha*), **P = 0.0063 (1 nM MP-10 vs. control; *Ucp1*), ****P < 0.0001 (100 nM MP-10 vs. control; *Ucp1*), ##P = 0.0017 (1 nM MP-10 vs. 100 nM MP-10; *Ucp1*), §§§§P < 0.0001 (10 μM cAMP vs. 100 nM MP-10; *Ucp1*), §§§§P < 0.0001 (10 μM cGMP vs. 100 nM MP-10; *Ucp1*), §§§§P < 0.0001 (10 μM cA + cG vs. 100 nM MP-10; *Ucp1*), *P = 0.0112 vs. (100 nM MP-10 vs. control; *Cidea*), ***P = 0.0007 (1 nM MP-10 vs. control; *Dio2*), ****P < 0.0001 (100 nM MP-10 vs. control; *Dio2*), #P = 0.0267 (1 nM MP-10 vs. 10 μM cAMP; *Dio2*), §§§P = 0.0001 (10 μM cAMP vs. 100 nM MP-10; *Dio2*), §§§P = 0.0007 (10 μM cGMP vs. 100 nM MP-10; *Dio2*), *P = 0.0232 (10 μM cA + cG vs. control; *Dio2*) §§P = 0.0058 (10 μM cA + cG vs. 100 nM MP-10; *Dio2*), *P = 0.0416 (10 μM cGMP vs. control; *Prdm16*), and *P = 0.0105 (10 μM cA + cG vs. control; *Prdm16*) using one-way analysis of variance (ANOVA) with Tukey's *post hoc* test.

B  RT–qPCR analysis of relative mRNA expression of thermogenic genes in cultured primary human white adipocytes in response to chronic treatments (8 h) with DMSO (control), MP-10 (1 and 100 nM), cyclic AMP (cAMP—10 μM), cyclic GMP (cGMP—10 μM), and cAMP + cGMP (cA + cG—10 μM (n = 4 separate cultures). *P = 0.0186 vs. (1 nM MP-10 vs. control; *Ucp1*), ****P < 0.0001 (100 nM MP-10 vs. control; *Ucp1*), ##P = 0.0016 (1 nM MP-10 vs. 100 nM MP-10; *Ucp1*), ***P = 0.0003 (10 μM cAMP vs. control; *Ucp1*), §§§P = 0.0002 (10 μM cGMP vs. 100 nM MP-10; *Ucp1*) and ***P = 0.0003 (cA + cG vs. control; *Ucp1*) using one-way analysis of variance (ANOVA) with Tukey's *post hoc* test.

Data information: Individual data from separate experiments are shown where bars represent mean ± SEM.

## Discussion

The PDE multifamily of isozymes have drawn considerable attention in the pharmaceutical industry over the past 30 years. Small-molecule inhibitors of the various PDEs provide effective treatments for a wide spectrum of diseases, including acute heart failure (milrinone), chronic obstructive pulmonary disease (roflumilast), erectile dysfunction (sildenafil, tadalafil, vardenafil), and pulmonary hypertension (sildenafil). This remarkable degree of success continues to spur interest in PDEs in various domains of biomedical research and in the development of new pharmacotherapies. Given its selective expression in striatum (Seeger *et al*, 2003; Coskran *et al*, 2006; Xie *et al*, 2006; Lakics *et al*, 2010; Jager *et al*, 2012), where it is the major cAMP-degrading enzyme (Russwurm *et al*, 2015) involved in regulating neuronal activity (Dedeurwaerdere *et al*, 2011; Piccart *et al*, 2014; Padovan-Neto *et al*, 2015; Wilson *et al*, 2015), plasticity (Yagishita *et al*, 2014), and gene expression (Siuciak *et al*, 2006; Kleiman *et al*, 2011), PDE10A has emerged as a prime candidate for the treatment of various psychiatric and neurodegenerative diseases (Wilson & Brandon, 2015).

In the present study, we discovered a novel thermoregulatory role for PDE10A in adipose tissue. We demonstrated that in addition to striatum, PDE10A is also expressed in interscapular BAT and peri-ovarian VAT in mice. Accordingly, acute inhibition of PDE10A with the highly selective inhibitor MP-10 elicited responses in striatum, BAT and VAT but not SAT and hypothalamus. Moreover, chronic MP-10 treatment caused significant weight reduction in DIO mice likely through elevated energy expenditure by browning of peri-ovarian VAT which was associated with enhanced insulin sensitivity. Furthermore, we demonstrated that PDE10A is markedly expressed in human supraclavicular BAT and abdominal SAT. Finally, the inhibition of PDE10A with MP-10 stimulated lipolysis and thermogenic gene expression in human brown adipocytes and induced browning of human white adipocytes.

The presence of metabolically active BAT in adult humans, coupled with the browning capacity of WAT, has rendered these thermogenic tissues important targets in the fight against obesity. Strategies are required other than classical stimulators of BAT that avoid ß-adrenoceptor-mediated cardiovascular and other adverse effects such as thyrotoxicosis. Moreover, an effective pharmacotherapy that selectively increases cAMP/cGMP signaling in adipose tissue has yet to be developed. Although the cAMP/cGMP dependency of adaptive thermogenesis has been extensively studied (Cannon & Nedergaard, 2004; Lafontan *et al*, 2008), the specific roles played by the negative regulators of cyclic nucleotide signaling have only relatively recently started to be addressed in this context.

In the present study, we have shown that PDE10A is an important regulator of at least three major processes in BAT: thermogenic gene expression, lipolysis, and glucose uptake. While acute administration of MP-10 to mice elevates cAMP levels in striatum (Grauer *et al*, 2009), this likely has minimal influence on BAT function (Zheng *et al*, 2013). Importantly, we demonstrated that acute administration of MP-10 has no effect on the expression of hypothalamic neuropeptides, which are known to recruit BAT via the sympathetic nervous system (Brito *et al*, 2007; Shi *et al*, 2013), in accordance with a lack of PDE10A expression in this brain region. In cell culture experiments, MP-10 stimulated lipolysis and induced thermogenic gene expression in both murine and human cells,

proving a cell-autonomous function of PDE10A in adipocytes. The stimulation of lipolysis by high-dose MP-10 was significantly higher than for either cAMP or cGMP treatments alone, but was indistinguishable from dual treatment with cAMP and cGMP. Therefore, it can be inferred that inhibiting PDE10A engages both cyclic nucleotide signaling arms in adipocytes in an additive fashion. Lipolysis is triggered in brown adipocytes by direct phosphorylation of lipid droplet-associated proteins such as perilipin and hormone-sensitive lipase by both PKA and PKG (Bordicchia *et al*, 2012), whereas thermogenic gene expression is regulated downstream of p38 MAP kinase, which is a common node of cAMP/PKA and cGMP/PKG signaling (Bordicchia *et al*, 2012). Glucose uptake by brown adipocytes in response to ß-adrenoceptor agonists requires intact mTORC2 signaling downstream of adenylate cyclase (Olsen *et al*, 2014; Albert *et al*, 2016). To date, glucose uptake by brown adipocytes downstream of PKG signaling has not been addressed, but may be mediated through AMPK (Deshmukh *et al*, 2010), the insulin receptor (Piwkowska *et al*, 2014), or PI3K (Doronzo *et al*, 2011) as has previously been shown for skeletal muscle, podocytes, and fibroblasts, respectively. The diminished uptake of FDG by BAT we found in DIO mice is consistent with observations of decreased FDG uptake in response to adrenergic stimuli in obese mice (Roberts-Toler *et al*, 2015), rats (Schade *et al*, 2015), and humans (Carey *et al*, 2013). Collectively, these results suggest a predominantly peripheral effect of MP-10-mediated inhibition of PDE10A on BAT function.

Classical BAT was originally deemed the principle thermogenic organ regulating non-shivering adaptive thermogenesis and body weight in mice (Lowell *et al*, 1993). More recently, browning of WAT has additionally been implicated in these homeostatic processes (Guerra *et al*, 1998; Schulz *et al*, 2013; Cohen *et al*, 2014). Studies in which PDEs are inactivated long-term illustrate the function and plasticity of these different fat depots. For instance, overexpression of microRNA-378 in adipose tissue of mice results in selective PDE1B inhibition in BAT and its subsequent expansion protecting from DIO (Pan *et al*, 2014). Chronic administration of the selective PDE4 inhibitor rolipram causes weight loss in DIO mice through browning of epididymal VAT (Park *et al*, 2012). Although less attention has been given to cGMP signaling, chronic administration of the selective PDE5 inhibitor sildenafil and small-molecule activators of soluble guanylate cyclase, which both elevate intracellular cGMP levels, has been shown to cause weight loss in DIO mice through browning of inguinal SAT (Mitschke *et al*, 2013; Hoffmann *et al*, 2015). The induction of thermogenic gene expression in peri-ovarian VAT in response to acute and chronic inhibition of PDE10A with MP-10 is consistent with previous studies in rats revealing that this depot in particular is susceptible to browning (Cousin *et al*, 1992). Indeed, acute treatment with a beta 3 adrenergic receptor agonist was sufficient to induce robust *Ucp1* mRNA expression in peri-ovarian VAT (Cousin *et al*, 1992). Notably, there appeared to be a counterintuitive reduction in UCP1 protein in BAT of mice chronically treated with MP-10. A similar reduction in UCP1 staining in BAT was also recently documented in mice that chronically received a synthetic thyroid hormone receptor agonist (Lin *et al*, 2015). This compound markedly stimulated thermogenesis, energy expenditure, and induced weight loss in ob/ob mice specifically through browning of SAT (Lin *et al*, 2015).

On the other hand, it has been suggested that increased PDE expression/activity in adipose tissue may underlie some forms of obesity. In an early study performed on pre-obese leptin-deficient Zucker rats, increased PDE2 expression and activity were found in BAT compared to lean heterozygotes (Coudray *et al*, 1999). In HF DIO mice, the pro-inflammatory enzymes IKKε and TBK1 are thought to increase PDE3B function in VAT and SAT through phosphorylation of Ser318 (as with Akt) contributing to catecholamine resistance (Mowers *et al*, 2013). In humans, however, an inverse correlation was found between total PDE and PDE3 enzyme activity in omental adipose tissue/adipocytes and BMI (Omar *et al*, 2011). Our findings now reveal increased PDE10A expression in BAT in DIO and ob/ob mice. As with PDE3B, this may also be due to increased inflammatory signaling associated with obesity as the PDE10A promotor harbors a binding site for the pro-inflammatory transcription factor NF-kappaB in mice and humans (Hu *et al*, 2004). Indeed, there seems to be a trend of increased PDE10A protein in omental and subcutaneous adipose tissue/adipocytes in obese compared to non-obese humans (Omar *et al*, 2011). Overall, these data suggest that the increase in expression/activity of a variety of PDEs in BAT and WAT, both in humans and animal models, results in decreased cAMP/cGMP signaling, promoting reduced thermogenesis, energy expenditure and obesity.

We found that pharmacological inhibition of PDE10A with MP-10 not only decreased its enzymatic function (as deduced by elevations in cAMP and cGMP concentrations in tissue), its mRNA expression levels were also selectively decreased both in cultured human brown adipocytes and in mouse tissues expressing the enzyme. This would suggest that cAMP and/or cGMP act to decrease *Pde10a* mRNA expression in a negative feedback fashion, which was, however, not supported by experiments performed with cyclic nucleotide treatments of human brown adipocytes. Negative feedback of PDE4D and PDE5 enzymatic function through PKA (Mika *et al*, 2015) and PKG (Gopal *et al*, 2001) has been described, and PDE10A itself is a cAMP-inhibited cGMP PDE (Fujishige *et al*, 1999). In contrast, with regard to mRNA expression, positive feedback has been reported for PDE7B in rat cultured striatal neurons (Sasaki *et al*, 2004) and PDE3 mRNA in cultured myoblasts (Kovala *et al*, 1994) through PKA activity. The mechanisms for reduced *Pde10a* mRNA expression in response to MP-10 treatment are unclear and require further study, but it is notable that MP-10 reduces both the catalytic activity and mRNA expression of PDE10A, which would serve to further amplify its effects.

There has been conflicting evidence concerning the influence of PDE10A inhibition on insulin secretion by pancreatic beta cells *in vitro* with an increase reported in one study (Cantin *et al*, 2007) and none reported in another (Nawrocki *et al*, 2014). At the organismal level however, improvements in glucose handling after treatment of mice with PDE10A inhibitors have consistently been found (Cantin *et al*, 2007; Nawrocki *et al*, 2014). We also observed improved insulin sensitivity in mice following chronic MP-10 treatment. Notably, such a robust effect of insulin on circulating glucose further corroborates MP-10 browning of WAT which is thought to disproportionately improve glucose handling in relation to body weight changes (Seale *et al*, 2011).

Chronic stimulation of supraclavicular BAT in humans by repeated cold exposure or administration of capsinoids has been shown to cause elevated energy expenditure and reduced adiposity (Yoneshiro *et al*, 2013). Here, we demonstrate for the first time high PDE10A expression in this region *in vivo*. This is consistent with a recent RNA sequence analysis study which revealed appreciable *Pde10a* mRNA expression in brown adipocytes derived from supraclavicular BAT (Xue *et al*, 2015). We also found marked expression of PDE10A in abdominal SAT consistent with a previous study showing marked PDE10A protein expression in both human abdominal SAT/adipocytes and omental VAT/adipocytes (Omar *et al*, 2011). The marked uptake of [$^{18}$F]-MNI-659 in the abdominal region in human scans in the present study could be additionally attributable to PDE10A binding to omental fat. In line with the findings of increased PDE10A expression in BAT of DIO and ob/ob mice, we found a similar pattern in patients with higher BMI. It should be noted, however, that with the limited sample size as well as the gender confound, it is difficult to draw any definitive conclusions. Nevertheless, chronic PDE10A inhibition could potentially recruit both BAT and BeAT resulting in increased energy expenditure with subsequent weight reduction in obese individuals.

At brain level, our small-animal PET/MRI studies with [$^{18}$F]-AQ28A confirmed high and specific expression of PDE10A in mouse striatum similar to other PDE10A-specific PET radioligands (Celen *et al*, 2010, 2013; Fan *et al*, 2014; Kehler *et al*, 2014; Ooms *et al*, 2014a; Harada *et al*, 2015; Toth *et al*, 2015). In line with previous studies, we found that acute MP-10 treatment induced *Fos*, *Zif-268*, and *Ppe* mRNA expression in striatum (Strick *et al*, 2010; Piccart *et al*, 2013). The neuropeptide precursor preprodynorphin/preprotachykinin and preproenkephalin are expressed in striatonigral/direct and striatopallidal/indirect pathway medium spiny neurons (MSNs) of the striatum, respectively (Lee *et al*, 1997). Although MP-10 has been shown to increase substance P (a cleavage product of preprotachykinin) expression previously (Strick *et al*, 2010), we did not find an effect on *Ppd* mRNA expression. This suggests that MP-10 predominantly engages indirect pathway D2 receptor-containing neurons, which is in agreement with a recent detailed Fos mapping study (Wilson *et al*, 2015).

As an important brain region involved in the motivational and reinforcing properties of natural rewards such as food, striatal function, particularly that of the dopamine system, has been shown to be dysregulated in obesity in preclinical models and humans (Horstmann *et al*, 2015). Previous studies on HF DIO and ob/ob mice have shown decreased dopamine release and/or dopamine receptor signaling in striatum (Fulton *et al*, 2006; Vucetic *et al*, 2012; Tellez *et al*, 2013). We here provide evidence that another molecule downstream of dopamine receptor signaling in the striatum is affected in DIO. The increased PDE10A levels we observed in striatum of DIO and particularly ob/ob mice may underlie increased feeding on a palatable diet, congruent with the decreased feeding on a HFD observed with PDE10A KO mice (Nawrocki *et al*, 2014).

A limitation of the present study is that we restricted our phenotypic analysis to female mice. We selected this gender as female PDE10A KO mice display a more marked phenotype than male counterparts and detailed metabolic studies have not been performed (Siuciak *et al*, 2006). Previous studies with the alternative PDE10A-specific inhibitor THPP-6 also showed weight loss and increased energy expenditure in male C57BL/6 mice (Nawrocki *et al*, 2014). Therefore, a sexually dimorphic and strain-specific effect of PDE10A in energy homeostasis is unlikely. It should be

noted, however, that browning of other WAT depots may occur in response to chronic PDE10A inhibition in male mice. Another limitation of the present study is that we did not assess PDE10A isoforms in adipocytes. PDE10A exists as two splice variants, PDE10A1 and PDE10A2, which differ in their amino terminus and in striatal neurons have been shown to be cytosolic and membrane bound, respectively (Charych *et al*, 2010). Interestingly, in the study of Kraynik *et al* (2013), treatment of brown adipocytes with the selective PDE 3 inhibitor cilostamide (with isoproterenol) induced *Ucp1* expression, while treatment with rolipram (alone) stimulated lipolysis, which suggests compartmentalization of function of PDEs in this cell type. Future studies can establish which splice variants of PDE10A are found in adipocytes and their respective subcellular distribution and roles in glucose uptake, lipolysis, and inducing gene expression. Our results further attest to the sensitivity of small-animal PET studies in measuring changes in PDE10A levels in preclinical models of disease such as has recently been shown in a Huntington's disease mouse model (Ooms *et al*, 2014b). It is noteworthy that changes in PDE10A expression may differ within sub-regions of the striatum in obesity. In a rat model of Parkinson's disease, it was shown that ventral striatal PDE10A expression and activity increase with concomitantly decreased dorsal striatal expression and activity (Giorgi *et al*, 2011). Future cellular/molecular studies can more accurately dissect the striatal sub-regions and pathways affected by altered PDE10A expression with DIO.

In summary, we have demonstrated an entirely novel thermoregulatory role for PDE10A in mouse and human adipocytes. Manipulations targeting other PDEs have no demonstrable beneficial effect on food intake (Ayala *et al*, 2007; Park *et al*, 2012; Mitschke *et al*, 2013; Pan *et al*, 2014; Hoffmann *et al*, 2015) with Sildenafil shown to actually increase feeding (Ayala *et al*, 2007) and PDE3 inhibitors postulated to do so by interfering with leptin action in hypothalamic feeding circuits (Degerman *et al*, 2011; Sahu, 2011). Therefore, PDE10A inhibitors are ideally placed for the treatment of obesity due to their dual ability to suppress hedonic feeding (Nawrocki *et al*, 2014) and increase energy expenditure. We also found improved insulin sensitivity after chronic MP-10 treatment further supporting the use of PDE10A inhibitors for the management of hyperglycemia. The restricted expression profile of PDE10A also confers less off-target effects of PDE10A inhibitors. The findings that PDE10A is induced in human colon tumor cells (Li *et al*, 2015) and that MP-10 halts tumor cell growth through PKG signaling and induces apoptosis (Lee *et al*, 2015), further add that PDE10A inhibitors may have multiple applications in obese patients with comorbid colorectal cancer.

# Materials and Methods

### Animals

Experiments were performed on adult (aged 12–24 weeks) female outbred CD1 and C57BL/6 mice (Taconic Biosciences, Cologne, Germany). All animals were group-housed in pathogen-free facilities at 22 ± 2°C on a 12-h light/dark cycle and had free access to water and standard chow diet (Sniff GmbH, Soest, Germany) unless otherwise stated.

### Chemicals

PF2545920 (MP-10) (Selleckchem, Munich, Germany) was dissolved in DMSO (Sigma-Aldrich Chemie GmbH, Taufkirchen, Germany) as a concentrated stock solution (40 mg/ml), and aliquots were stored at −20°C. On the day of *in vivo* experimentation, an aliquot of stock solution was freshly diluted by a factor of 4 (and a factor of 12 for the chronic study) in Miglyol (Fagron GmbH & Co., Rotterdam, Holland) such that mice received a volume of 3 μl/g body weight. Vehicle solution was administered at the same volume and contained the same proportion of DMSO and Miglyol as MP-10 solution.

### *In vivo* phenotypic characterization

Six sets of mice were used for these studies:
Set 1 comprised normal weight CD1 mice ($n = 13$) used for small-animal PET-MRI scanning. The first subset ($n = 5$) received the PDE10A selective PET radioligand [18F]-AQ28A. The second subset ($n = 8$) underwent two [18F]-FDG-PET scans separated by an interval of 1 week; the first taking place after an overnight fast and the second in the *ad libitum* fed state. Although [18F]-FDG-PET scans are normally performed in the fasted state to avoid competition of radioligand with endogenous glucose, we opted for this protocol as fed mice exhibit higher [18F]-FDG uptake by BAT than fasted mice due to increased sympathetic tone (Fueger *et al*, 2006). After a final 1-week recovery period, *ad libitum* fed animals received MP-10 (30 mg/kg, $n = 7$) or vehicle control ($n = 6$) intraperitoneally (i.p.) in the early light phase. Two hours later, animals were anesthetized with isoflurane and sacrificed by intracardial perfusion of ice-cold PBS. White adipose tissue (inguinal and peri-ovarian) and BAT (interscapular) were then immediately excised and rapidly snap-frozen in liquid nitrogen. Brain was also immediately collected, partially frozen, and placed on its dorsal surface. Entire hypothalamic and striatal tissue blocks were then cut with a fine razor and subsequently snap-frozen in liquid nitrogen. Samples were stored at −80°C until subsequent analyses.
Set 2 comprised normal weight CD1 mice ($n = 12$) used for a cold challenge study and *ex vivo* cyclic nucleotide measurements. Animals were transferred to an incubator (HPP110 life, Memmert GmbH & Co. KG, Schwabach, Germany) set at 8°C for 4 h immediately following a single i.p. treatment with MP-10 (30 mg/kg, $n = 4$) or vehicle ($n = 4$). Core body temperature was measured hourly with a rectal probe (TH-5, Thermalert Monitoring Thermometer, Clifton, NJ, USA). After a 1-week recovery period, animals received a single i.p. treatment with either MP-10 (30 mg/kg, $n = 6$) or vehicle control ($n = 6$) before they were dissected 30 min later as described above.
Set 3 comprised normal weight CD1 mice rendered obese through *ad libitum* access to a high–fat, high sugar (HFHS) diet (Sniff GmbH) for 16 weeks ($n = 4$) before undergoing small-animal PET-MRI scanning with [18F]-AQ28A and [18F]-FDG (in the *ad libitum* fed state).
Set 4 comprised genetically obese leptin-deficient (ob/ob) female mice on a C57BL/6 background used for PET scanning ($n = 4$). These mice were maintained on a standard chow diet (Sniff GmbH) for 16 weeks and underwent small-animal PET-MRI scanning with the PDE10A tracer in the *ad libitum* fed state.

Set 5 comprised normal weight CD1 mice rendered obese (mean body weight, 41.8 ± 2.4 g, *n* = 8) through 16-week *ad libitum* access to HFHS diet (Sniff GmbH) used for metabolic phenotype characterization. Animals were singly housed and prepared via daily sham injections. During the 7-day feeding study, animals received daily i.p. administration of vehicle (*n* = 4) or 10 mg/kg MP-10 (*n* = 4) at the early light phase, during which body weight and food intake were measured. The day after completion of the chronic feeding study, animals were placed in metabolic cages, and after a 24-h acclimatization period, indirect calorimetry was assessed by a Calorimetry Module (TSE Systems, Bad Homburg, Germany) as previously described (Hesselbarth *et al*, 2015) under ongoing once-daily treatment with vehicle (*n* = 6) or 30 mg/kg MP-10 (*n* = 6). Following a 24-h recording period, animals received terminal i.p. injections of vehicle (*n* = 4) or 30 mg/kg MP-10 (*n* = 4) and tissues were dissected 2 h later as described above.

Set 6 comprised normal weight C57BL6 mice rendered obese (mean body weight 32.1 ± 0.89 g, *n* = 8) through 12-week *ad libitum* access to a HF diet (60 kcal% fat, D12492, Sniff GmbH) used for further metabolic phenotype characterization. Animals underwent a chronic feeding study as described above. The day after completion of the chronic feeding study, animals then underwent an insulin tolerance test. Baseline blood glucose values were measured in *ad libitum* fed animals and then after i.p. injection of 0.75U insulin, at 15, 30, 45, and 60 min. Animals were then sacrificed and fat was dissected as described above with the addition that extra samples were collected and immediately fixed in 4% PFA for subsequent histological analysis.

### Radiochemistry [18F]-AQ28A

[18F]-AQ28A has previously been demonstrated to be a highly specific radioligand of PDE10A. [18F]-AQ28A was produced as previously described (Wagner *et al*, 2015) by a fully automated synthesis procedure performed in a TRACERlabTM FX F-N platform (GE Healthcare, Frankfurt am Main, Germany). In brief, a nitro precursor was used for nucleophilic aromatic radiofluorination, and [18F]-AQ28A was isolated by semi-preparative HPLC followed by purification using a Sep Pak® C18 Plus light cartridge. [18F]-AQ28A was finally formulated in isotonic saline containing 10% ethanol. Chemical and radiochemical purities were equal to or greater than 97% with specific activities of 96–145 GBq/μmol at the end of a 70-min synthesis period.

### PET/MRI imaging

To avoid the confounding influence of diurnal variations in glucose uptake by BAT (van der Veen *et al*, 2012), all PET/MRI scans were performed during the light phase using a dedicated preclinical PET/MRI system (nanoScan®, Mediso Medical Imaging Systems, Budapest, Hungary) as previously described (Nagy *et al*, 2013; Gnad *et al*, 2014). For [18F]-FDG experiments, mice were housed overnight under thermoneutral conditions (29°C) in an incubator (HPP110 life, Memmert GmbH & Co. KG, Schwabach, Germany). Mice received intraperitoneal (i.p.) injections of either MP-10 (30 mg/kg) or vehicle control and were anesthetized (Anaesthesia Unit U-410, agntho's, Lidingö, Sweden) 20 min later with isoflurane

(1.8%, 0.35 l/min) delivered in a 60% oxygen/40% air mixture (Gas Blender 100 Series, MCQ instruments, Rome, Italy). Mice then received a bolus intravenous (i.v.) injection of 13.6 ± 2.4 MBq [18F]-FDG (supplier: Department of Nuclear Medicine, University of Leipzig, Germany) via the lateral tail vein timed 30 min after initial treatments. Simultaneous to tracer injection, a dynamic 55-min PET scan was initiated, during which animals were maintained at 37°C with a thermal bed system under isoflurane anesthesia. The list-mode data were reconstructed as 11 frames (11 × 5 min) with 3D-ordered subset expectation maximization (OSEM), 4 iterations, and 6 subsets using an energy window of 400–600 keV, coincidence mode of 1–5 and ring difference of 81.

For experiments with [18F]-AQ28A, animals were immediately anesthetized and PET-scanned after intravenous injection of 11.3 ± 2.7 MBq as described above. The list-mode data here were acquired as 23 frames (15 × 1 min and 8 × 5 min). After completion of PET scanning, a structural MRI scan (used for anatomical orientation and attenuation correction purposes) was performed on a 1T magnet using a $T_1$-weighted gradient echo sequence ($T_R$ = 15 ms, $T_E$ = 2.6 ms). For analysis of [18F]-FDG and [18F]-AQ28A uptake, the mean standardized uptake value (SUV) was calculated. All regions of interest (striatum, hypothalamus, interscapular BAT, and muscle) were located manually with reference to structural MRI scans after co-registration of dual-modality image data sets using ROVER software (ABX, v.2.1.17, Radeberg, Germany).

### RNA isolation and quantitative real-time PCR analysis

Total RNA was extracted using QIAzol Lysis Reagent and an RNeasy Mini Kit (Qiagen GmbH, Hilden, Germany) and 1 μg was reverse-transcribed using a QuantiTect RT Kit (Qiagen GmbH) at 42°C for 25 min followed by a 3-min 93°C denaturation step. 100 ng cDNA was added to qPCRs using a QuantiTect SYBR-Green Kit (Qiagen GmbH) and a Light Cycler 480 Instrument (Roche Diagnostics, Mannheim, Germany) under the following settings: incubation at 95°C for 15 min; amplification at 95°C for 5 s followed by 60°C for 1 min (40 cycles); melting at 95°C for 30 s, 60°C for 2 min, 45°C for 30 s and 95°C; and cooling at 4°C for 5 min. The sequences for primers used (Eurofins Genomics, Ebersberg, Germany) are shown in Appendix Table 1 and specificity of PCR products ensured with reference to melting curve data. The $\Delta\Delta C_T$ method was employed to calculate mRNA expression of the gene of interest relative to beta actin, and data are expressed relative to control group.

### Cyclic nucleotide determination

Cyclic nucleotide (cAMP and cGMP) concentrations in adipose tissue depots were determined by a competitive colorimetric ELISA (Abcam plc) according to the manufacturers' protocol. Briefly, tissue samples (50–100 mg) were pulverized in liquid nitrogen and suspended in 20 μl/mg 0.1 M HCl. Acetylating reagent (10 μl per 200 μl crude homogenate) was then added. A second finer homogenization step was applied using Precellys® 1.4-mm ceramic beads and homogenizer (Peqlab, Erlangen, Germany). Fine homogenate was then centrifuged at 44,800 *g* at 4°C for 15 min. Supernatant was collected and then transferred to a 96-well plate read at ODs of 405 and 580 nm. Reactions were performed in duplicate. Cyclic nucleotide concentrations were determined by interpolating from a

standard curve using GraphPad Prism version 5 (GraphPad Software Inc, La Jolla, USA) and then normalized to protein concentration and expressed as fmol/µg protein.

## Histology

Adipose tissues were harvested, fixed in 4% paraformaldehyde (pH 7.4) for 24 h at 4°C, and rinsed with phosphate-buffered saline before embedding in paraffin and sectioning at 5 µM. For UCP1 immunohistochemistry, rabbit anti-UCP1 polyclonal antibody (ab23841; Abcam, Cambridge, UK) was used. Sections were rinsed thoroughly and incubated with HRP-conjugated goat anti-rabbit immunoglobulin polyclonal antibody (P 044; Dako Envision™+; Dako, Hamburg, Germany) for 1 h. Visualization was achieved with 3,3′-diaminobenzidine. Microscopic examination was performed using an AxioObserver Microscope (Carl Zeiss, Jena, Germany) at a magnification of 10×. Images were obtained using ZEN2012 software (Carl Zeiss).

## Primary human and murine adipocyte culture

Primary human brown adipocytes (hMADs) were provided by the laboratory of Dr C. Dani (University of Nice Sophia Antipolis) and were differentiated as previously described (Hoffmann et al, 2015). Human myocytes and human white adipocytes (Promo Cell Bioscience, GmbH, Heidelberg, Germany) were differentiated according to the manufacturer's instructions. Murine preadipocytes were isolated from interscapular BAT of newborn mice and differentiated as previously described (Hoffmann et al, 2015). For lipolysis assays, differentiated human brown and murine brown adipocytes were washed twice with phenol-red-free DMEM (Life Technologies GmbH, Darmstadt, Germany) supplemented with 2% w/v fatty acid-free BSA (Sigma-Aldrich, Chemie GmbH) followed by incubation with lipolysis medium containing indicated substances for 4 h (primary human brown adipocytes) or 2 h (murine brown adipocytes), respectively. Cell culture media was collected, free glycerol reagent (Sigma-Aldrich, Chemie GmbH) was added, and free glycerol was measured at 540 nM. Total glycerol content was calculated with glycerol standard (Sigma-Aldrich) and normalized to protein content. For gene expression analysis, differentiated human brown and white adipocytes were treated as above and incubated with indicated substances for 8 h prior to RT–qPCR as described recently (Hoffmann et al, 2015). For experiments determining the effect of PDE10A inhibition on adipocyte differentiation as well as thermogenic markers in hMADS, indicated substances were applied to human adipocytes every second day during the differentiation process after growth and induction were stimulated.

## Retrospective analysis of human whole-body PET images with the PDE10A radioligand [18F]-MNI-659

A retrospective analysis of data from human whole-body PET dosimetry studies with the PDE10A radioligand [18F]-MNI-659 was performed (Barret et al, 2014). The demographics of the subjects studied can be found in Appendix Table 2. In the absence of structural CT or MRI images, we resorted to the attenuation correction map generated by the PET transmission scan for rough anatomical reference. The mean SUV of [18F]-MNI-659 was calculated in abdominal subcutaneous WAT, supraclavicular BAT, and skeletal muscle.

### The paper explained

#### Problem

Brown adipose tissue (BAT) and beige adipose tissue (BeAT) have emerged as attractive novel targets for obesity therapies due to their energy-expending properties. Drugs which recruit these tissues to stimulate thermogenesis without causing unwanted negative side effects are lacking.

#### Results

Here, we present evidence that phosphodiesterase type 10A (PDE10A) is expressed in murine and human white adipose tissue (WAT) and BAT. We demonstrate that acute pharmacological inhibition of PDE10A with MP-10, a highly selective and well-tolerated compound, sufficiently elicits multiple functional responses in WAT and BAT. We found a pattern of increased PDE10A expression in BAT of obese mice and humans and that chronic inhibition of PDE10A with MP-10 results in weight loss in diet-induced obese mice associated with increased energy expenditure, browning of visceral WAT and enhanced insulin sensitivity. We further demonstrate that MP-10 treatment stimulates thermogenic gene expression in primary human brown adipocytes and causes browning of primary human white adipocytes.

#### Impact

We have characterized an entirely novel role for PDE10A in adipose tissue and, for the first time, have translated the anti-obesity potential of phosphodiesterase inhibitors from preclinical models to humans. Our findings suggest that PDE10A inhibitors represent promising and safe weight loss compounds that improve glucose homeostasis in addition to potential treatments for various neurological conditions and certain cancers.

## Statistics

Sample sizes were determined according to our own previous studies and to published reports in which similar exploratory experimental procedures were performed. Investigators were not blinded to the studies performed in this manuscript. All data are presented as means ± SEM. Normality of distribution was ensured using the Kolmogorov–Smirnov test, and data sets were analyzed for statistical significance using unpaired 2-tailed Student's $t$-tests, one-way ANOVA corrected for multiple comparisons with Tukey's post hoc test, and two-way ANOVA corrected for multiple comparisons with Sidak's post hoc test with GraphPad Prism version 5 (GraphPad Software Inc). $P$-values < 0.05 were considered significant.

## Study approval

All animal experiments were approved by the local Animal Care and Use Committees of the State of Saxony, Germany, as recommended by the animal ethics review board (Regional Administrative Authority Leipzig, TVV 08/13 and TVV 63/13, Germany) and followed the NIH guidelines for care and use of animals. The study involving analysis of human [18F]-MNI-659 PET scans was reviewed and approved by the New England Institutional Review Board (IRB). All participating subjects of the original study (Barret et al, 2014) provided written informed consent. Since the present study was a retrospective analysis of existing data, the IRB granted a waiver of informed consent for this purpose.

Appropriate safeguards were in place to protect subject confidentiality and privacy.

**Expanded View** for this article is available online.

## Acknowledgements

We thank Dr. Mathias Kern, Nico Hesselbarth, Annett Hoffmann, Anja Mohl, Patricia Zehner, and Daniela Hass for excellent technical assistance with the studies described in the manuscript. We thank Dr. Barbara Wenzel, Dr. Rodrigo Teodoro, and Dr. Steffen Fischer for their kind assistance with the synthesis and development of [$^{18}$F]-AQ28A used in the small-animal PET-MRI studies performed in this manuscript. We also thank Dr. Stephen Smith, Dr. Zachary Gerhart-Hines, and Prof Mathias Faßhauer for advice and valuable comments on the manuscript. This work was supported by the Federal Ministry of Education and Research (BMBF), Germany, grant number FKZ: 01EO1501 for WKF and NK and Deutsche Forschungsgemeinschaft (DFG) AOBJ: 624808 for WKF (FE 1159/2-1), SFB 1052 Project B01 for MB, SFB 1052 B04 for NK and SFB 1052 Project C07 for JTH.

## Author contributions

MKH and WKF conducted the majority of the experiments. MKH and WKF wrote the manuscript. MK and WD-C performed and analyzed the small-animal PET/MRI studies. TG and AP performed the cell culture experiments. JW and JTH performed histology. FB assisted with the chronic feeding study and the RT–qPCR experiments. NK and MB performed cold exposure and metabolic energy expenditure tests. SW and PB synthesized and provided the [$^{18}$F]-AQ28A tracer for PET scans. JPS provided the human [$^{18}$F]-MNI-659 PET scan data for retrospective analysis, which was analyzed by KS, JL, SH, and OS, MKH and WKF conceived the project. MKH, MB, and WKF supervised the project. All authors discussed the results and commented on the manuscript .

## Conflict of interest

The authors declare that they have no conflict of interest.

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
