## [Review Process File · EMBO Molecular Medicine]

A novel thermoregulatory role for PDE10A in mouse and human adipocytes

Mohammed K Hankir, Mathias Kranz, Thorsten Gnad, Juliane Weiner, Sally Wagner, Winnie Deuther-Conrad, Felix Bronisch, Karen Steinhoff, Julia Luthardt, Nora Klüting, Swen Hesse, John P Seibyl, Osama Sabri, John T Heiker, Matthias Blüher, Alexander Pfeifer, Peter Brust and Wiebke K Fenske

Corresponding author: Wiebke K Fenske, IFB Adiposity Diseases University of Leipzig

Review timeline:	Submission date:	21 November 2015
	Editorial Decision:	16 December 2015
	Revision received:	18 March 2016
	Accepted:	25 April 2016

Transaction Report:

Editor: Céline Carret

1st Editorial Decision

16 December 2015

Thank you for the submission of your manuscript to EMBO Molecular Medicine. We have now heard back from the two referees whom we asked to evaluate your manuscript. Although the referees find the study to be of potential interest, they also raise a number of concerns that should be addressed in the next final version of your article.

As you will see, both referees are rather positive about the study but while referee 2 is overall more supportive, referee 1 suggests a number of additional but reasonable and feasible, experiments that would definitely in our view increase the conclusiveness of the findings as well as the translational relevance (a point shared by both referees). I would thereby encourage you to perform all suggested experiments, including detailing the pharmacological use of MP-10.

Given the balance of these evaluations, we feel that we can consider a revision of your manuscript if you can address the issues that have been raised within the 3-months constraints of a revision time. Please note that it is EMBO Molecular Medicine policy to allow only a single round of major revision and that, as acceptance or rejection of the manuscript will depend on another round of review, your responses should be as complete as possible.

Please also contact us as soon as possible if similar work is published elsewhere. If other work is

published we may not be able to extend the revision period beyond three months.

Please read below for important editorial formatting.

I look forward to receiving your revised manuscript.

***** Reviewer's comments *****

Referee #1 (Remarks):

Comments to the manuscript: EMM-2015-06085

The manuscript entitled "A novel thermoregulatory role of PDE10A in mouse and human adipocytes" by Hankir et al., investigated the potential role of phosphodiesterase type 10A (PDE10A) as new factor involved in thermoregulation both in mouse and human adipocytes. Moreover they identified inhibitors of PDE10A activity as promising candidates for the treatment of human obesity.

The novelty of the manuscript is in the identification of PDE10A as a thermoregulatory factor which inhibition promotes direct brown adipose tissue activation and visceral white adipose tissue browning.

The manuscript is well written making it easy to read. However, few more experiment and validations are requested for their results to support completely their conclusions.

Major comments

Methods

-The authors use the CD1 outbred mouse strain for their experiments concerning diet induced obesity and thermogenesis, instead of the more widely used inbred C57Bl6? The authors should give a rationale concerning their choice.

-Moreover, the authors for their diet induced obesity experiment use a high fat high carbohydrate diet instead of only a high fat diet? Can the authors explain why?

-The authors should precise the age of "adult" female used in the study.

-The duration of the MP-10 treatment in vivo and in vitro should also appear in the method section.

Results:

Figure 1.

How specific is the radio-ligand [18F]-AQ28A for PDE10A with respect to other phosphodiesterases? The authors should provide evidence that this ligand bind only to PDE10A and not to other isoenzymes.

In Figure 1, the authors should include a PET/MRI/PET+MRI image of the BAT of the mice before [18F]-AQ28A injection, as a negative control. The same for the hypothalamus/striatum image in Supplementary Figure 1.

Representative images should also include the brain. What is the impact of cold exposure on [18F]-AQ28A uptake in BAT?

Figure 2.

A. What is difference between the PDE10A inhibitor THPP-6 previously validated in vivo (Nawrocki 2014) and MP-10? How selective is the PDE10A inhibitor MP-10? How the authors chose the working dose? To validate the inhibition of PDE10A by MP-10 the authors should provide evidence of decreased activity or expression (mRNA and/or protein levels) of the enzyme in the BAT of the mice. Moreover they should provide also data validating the specificity of MP-10 for PDE10A looking at the activity/expression of other isoenzymes in the mouse BAT after the

administration of the compound.

Given that several phosphodiesterase are co-expressed in AT including PDE3B, the authors should compare the gene expression levels in brain, adipose tissues and muscle, in order to confirm the physiological relevance of PDE10A.

B. The authors made the interesting observation that mice treated with MP-10 were protected from hyperthermia. Authors should validate increased BAT activation by measuring [18F]-FDG uptake in BAT.

C. The authors affirmed PDE10A is mainly expressed in brain, BAT and in a lesser extent in muscle. Which cells are expressing PDE10A? Given that BAT progenitors share common markers with muscle progenitors, it would be interesting to figure out the gene expression levels in BAT progenitors compared to other cell types?

E. Data depicted in Figure 2E are interesting and convincing, however the authors should confirm VAT browning at the protein/cellular levels through IHC analysis (e.g. UCP-1 protein expression, locularity, adipocytes cell size, etc...). The authors should confirm also for this panel that the increase in browning in visceral WAT vs SAT and increased expression of thermogenic markers in BAT are a consequence of a specific decrease in PDE10A expression and not also of other isoenzymes following the treatment with MP-10. Same for the striatum in Supplementary Figure 3C.

How the authors can differentiate the physiological effects attributed to BAT/VAT or other tissues like the brain? The authors should investigate the impact of PD-10A in the brain through measuring cAMP levels and showed locomotor activity (Figure 5E) earlier in the manuscript.

Figure 3.

C. Once again, also for the human MADS experimental model treated chronically with MP-10, the authors should provide evidence that is the specific decreased activity and/or expression of PDE10A that causes browning of these cells.

-The *in vitro* experiments depicted in Figure 3 are incomplete since the authors should also investigate the effect of PDE10A inhibition on differentiation of primary brown and white cells (VAT depots).

-What about lipolytic activity *in vivo*, in mice receiving MP-10 inhibitors? Did authors measure NEFA circulating levels?

Figure 4.

A. What about the uptake of the radiolabelled ligand in VAT vs SAT? If the specific inhibition of PDE10A induces browning in visceral WAT vs SAT, we would expect an increase of uptake of [18F]-AQ28A in visceral WAT vs SAT in DIO vs lean mice. It would be interesting to have these data included in the manuscript.

Figure 5.

A-B. why the authors changed the concentration of the inhibitor MP-10 from 30 to 10mg/kg? Moreover, how the authors explain that even vehicle-treated DIO mice are losing weight? Are these mice still in high fat diet?

Metabolic phenotyping is incomplete. Authors should investigate the effect of MP-10-induced weight loss/increased energy expenditure on insulin sensitivity.

C. The Resting Energy Expenditure in this case should not be corrected by body weight. The mice treated with MP-10 are leaner with respect to the control mice, so forcedly the EE of these mice will result decreased vs untreated. To confirm that the decreased EE is not an artefact of the calculation, the EE should be normalised to the lean mass or assuming that the mice weight the same.

F. On top of thermogenic and beige markers the authors should measure in the SAT, VAT and BAT of the mice also the expression of PDE10A to validate the effect of MP-10.

Once again, the authors should confirm VAT browning through histology analyses.

Figure 6.

The authors should provide also a representative PET image of before the human subject received the [18F]-MNI-1869 ligand as positive control. Moreover they should also confirm the specificity of this ligand [18F]-MNI-1869 for human PDE10A. The authors should add on the graph the [18F]-MNI-1869 uptake in the brain of human subjects. It would be also interesting to plot individual

curves of [¹⁸F]-MNI-1869 uptake. Considering the fact that 2 of these patients are obese or overweight, it would be interesting to see if they display higher uptake of PDE10A radioligand.

Minor comments

Please correct the typo page 7 "...the adoption of BAT phenotype..."

Results:

Figure 2.

A. Why the authors perform the experiment both a fasted and ad libitum fed state? And not only a fed state?

In the second part of the paragraph entitled "PDE10A regulates brown and visceral white adipose tissue thermogenic function", the numbering of Figure 2 and Supplementary Figure 2 panels in the text with respect to the figures themselves is wrong. We ask the authors to check that carefully.

Referee #2 (Remarks):

As stated by the authors, the current study describes an entirely novel role for PDE10A in adipose tissue and further support that selective PDE10A inhibitors may represent ideal treatment candidates for the management of obesity due to their dual ability to suppress feeding and increase energy expenditure.

Based on the novelty of these data and the fact that they were able to show that expression and function of PDE10A in adipose tissue translated from mouse to humans, I believe this manuscript will have a high impact on the field.

My only concerns stem from the lack of strong controls for the imaging studies. While the wealth of data argues that the biological effects they characterize in this study were driven by PDE10A, I believe the authors should have used PDE10A knockout mice to confirm their findings. A second concern is that the investigators used a very high dose of the PDE10A inhibitor MP-10 and it would be useful to run a dose titration of this compound to understand the PDE10A occupancy / efficacy relationship in the model system.

However, I feel the data in the manuscript still warrants publication without further studies. I suggest the authors implement my suggestions in future publication exploring this exciting new mechanism

1st Revision - authors' response

18 March 2016

Referee #1:

Major comments

Methods

R1. The authors use the CD1 outbred mouse strain for their experiments concerning diet induced obesity and thermogenesis, instead of the more widely used inbred C57Bl6? The authors should give a rationale concerning their choice.

A1. We chose CD1 mice as this strain was used for our previous studies utilizing [¹⁸F]-AQ28A. We have however performed additional metabolic studies on C57BL/6 mice confirming the results found in the CD1 strain (Figure 4 and Supplementary Figure 9).

R2. Moreover, the authors for their diet induced obesity experiment use a high fat high carbohydrate diet instead of only a high fat diet? Can the authors explain why?

A2. We opted for a high fat high sugar diet as we have found this to cause DIO more rapidly. We did however perform additional metabolic studies on high fat DIO C57BL/6 mice (Figure 4 and Supplementary Figure 9).

R3. The authors should precise the age of "adult" female used in the study.

A3. We have now indicated the age of our mice in the methods section (line 491).

R4. The duration of the MP-10 treatment in vivo and in vitro should also appear in the method section.

A4. We have now indicated the duration of MP-10 treatments in the methods section (lines 533-540, 626-631 and 634-636).

Results:

Figure 1

R5. Figure 1. How specific is the radio-ligand [18F]-AQ28A for PDE10A with respect to other phosphodiesterases? The authors should provide evidence that this ligand binds only to PDE10A and not to other isoenzymes.

A5. We have data in our other manuscript, which is available online now (Wagner et al., Eur J Med Chem, 2016) that [18F]-AQ28A is a highly specific radioligand to PDE10A with an IC₅₀ value of 2.91 nM (compared to values > 1000 nM of all other PDE isoenzymes) (for further details, see table 4 in the above mentioned manuscript) (lines 555-556). We have also performed a blocking experiment with MP-10 in which we observed displacement of [18F]-AQ28A in BAT and striatum indicative of its specific binding to PDE10A (Supplementary Figure 2).

R6. In Figure 1, the authors should include a PET/MRI/PET+MRI image of the BAT of the mice before [18F]-AQ28A injection, as a negative control. The same for the hypothalamus/striatum image in Supplementary Figure 1.

A6. Unfortunately, we cannot add a PET/MRI/PET+MRI image without administration of radiotracer as according to the concept of this imaging technique and standard practice, PET scans started with the onset of radiotracer administration (unlike for the use of contrast agents in MRI or CT).

R7. Representative images should also include the brain. What is the impact of cold exposure on [18F]-AQ28A uptake in BAT?

A7. We have now added corresponding MRI and fused PET/MRI images of the brain in Supplementary Figure 1. We did not test the effects of cold exposure on [18F]-AQ28A uptake in BAT but these would be interesting future follow-up experiments.

Figure 2.

R8. What is difference between the PDE10A inhibitor THPP-6 previously validated in vivo (Nawrocki 2014) and MP-10? How selective is the PDE10A inhibitor MP-10? How the authors chose the working dose?

A8. MP-10 is a far more commonly used and established PDE10A inhibitor than the novel Merck compound THPP-6 (which is not yet commercially available). Previous data in vitro have shown that MP-10 has greater than 1,000-fold selectivity for PDE10A over other PDEs (Grauer et al., 2009 The Journal of Pharmacology and Experimental Therapeutics) compared to a 300-fold selectivity of THPP-6 over the other 10 PDE isozymes (Nawrocki 2014, table 2). The working dose of 30mg/kg of MP-10 was selected on the basis that it fell within the range based on previous studies performed on mice revealing robust effects on striatal glucose metabolism (Dedeurwaerdere et al., 2011 The Journal of Pharmacology and Experimental Therapeutics) which we aimed to initially reproduce for BAT glucose metabolism.

R9. To validate the inhibition of PDE10A by MP-10 the authors should provide evidence of decreased activity or expression (mRNA and/or protein levels) of the enzyme in the BAT of the mice. Moreover they should provide also data validating the specificity of MP-10 for PDE10A looking at the activity/expression of other isoenzymes in the mouse BAT after the administration of the compound.

A9. We agree with the reviewer and measured in additional experiments Pde10a and Pde3b mRNA expression in SAT, VAT, BAT and striatum after acute MP-10 treatment (30mg/kg). We did indeed find selective reduction of Pde10a mRNA in VAT, BAT and striatum (Supplementary Figure 7 in the revision).

R10. Given that a several phosphodiesterase are co-expressed in AT including PDE3B, the authors should compared the gene expression levels in brain, adipose tissues and muscle, in order to confirm the physiological relevance of PDE10A.

A10. We previously performed a distribution analysis of Pde3b and Pde10a mRNA in mice. The distribution of Pde10a in adipose tissue was consistent with the effects on cyclic nucleotide concentrations and gene expression in response to acute inhibition of PDE10A with MP-10 treatment (Figure 2C-E). In brain, the expression profile of Pde10a was again consistent with the effects on gene expression in response to acute inhibition of PDE10A with MP-10 treatment (Supplementary Figure 6).

R11. Figure 2B. The authors made the interesting observation that mice treated with MP-10 were protected from hyperthermia. Authors should validate increased BAT activation by measuring [18F]-FDG uptake in BAT.

A11. We performed a comprehensive set of FDG PET experiments evaluating FDG uptake by BAT in response to acute inhibition of PDE10A with MP-10 (Figure 2A and Supplementary Figure 3).

R12. Figure 2C. The authors affirmed PDE10A is mainly expressed in brain, BAT and in a lesser extent in muscle. Which cells are expressing PDE10A? Given that BAT progenitors share common markers with muscle progenitors, it would be interesting to figure out the gene expression levels in BAT progenitors compared to other cell types?

A12. We agree with the reviewer and measured in additional experiments now Pde10a and Pde3b mRNA expression in human brown adipocyte, human white adipocyte and human myocyte precursors (Supplementary Figure 12).

R13. Data depicted in Figure 2E are interesting and convincing, however the authors should confirm VAT browning at the protein/cellular levels through IHC analysis (e.g. UCP-1 protein expression, locularity, adipocytes cell size, etc...).

A13. We reserved IHC analysis for animals receiving chronic treatment of MP-10 (see below A.23).

R14. The authors should confirm also for this panel that the increase in browning in visceral WAT vs SAT and increased expression of thermogenic markers in BAT are a consequence of a specific decrease in PDE10A expression and not also of other isoenzymes following the treatment with MP-10. Same for the striatum in Supplementary Figure 3C.

A14. As mentioned above, we found that acute treatment of mice with MP-10 selectively reduced Pde10a mRNA expression in striatum, BAT and VAT with no effects on Pde3b mRNA expression (Supplementary Figure 7).

R15. How the authors can differentiates the physiological effects attributed to BAT/VAT or other tissues like the brain? The authors should investigate the impact of PDE-10A in the brain through measuring cAMP levels and showed locomotor activity (Figure5E) earlier in the manuscript.

A15. To rule out a CNS mediated effect of MP-10, we additionally measured gene expression in hypothalamus and striatum which is well known to regulate BAT. Indeed we found no effects on a

panel of hypothalamic neuropeptides (Supplementary Figure 6). MP-10 has previously been shown to increase cyclic nucleotide levels in mouse striatum (Grauer et al., 2009 *Journal of Pharmacology and Experimental Therapeutics*). However, a study in which PKA in striatum is dysregulated in the long-term (Zeng et al., 2013 *PNAS*) has shown that this unlikely has an influence on BAT function. In Figure 5F, in which we measure locomotor activity after MP-10 treatment, we did not register an effect.

Figure 3.

R16. Figure 3C. Once again, also for the human MADS experimental model treated chronically with MP-10, the authors should provide evidence that is the specific decreased activity and/or expression of PDE10A that causes browning of these cells.

A16. In additional experiments, we found that acute treatment of human brown adipocytes (hMADs) selectively reduced Pde10a mRNA expression with no effect on Pde3b expression (Supplementary Figure 7). This is entirely consistent with the data obtained from the mouse studies.

R17. The in vitro experiments depicted in Figure 3 are incomplete since the authors should also investigate the effect of PDE10A inhibition on differentiation of primary brown and white cells (VAT depots).

A17. In response to this request, we additionally investigated the effects of PDE10A inhibition on the differentiation of primary brown and white human adipocytes (Supplementary Figure 12).

R18. What about lipolytic activity in vivo, in mice receiving MP-10 inhibitors? Did authors measure NEFA circulating levels?

A18. We unfortunately did not collect plasma samples from our acute studies with MP-10, but we agree that this analysis would nicely complement our in vivo findings of MP-10 effects on thermogenesis in future studies.

Figure 4.

R19. Figure 4A. What about the uptake of the radiolabelled ligand in VAT vs SAT? If the specific inhibition of PDE10A induces browning in visceral WAT vs SAT, we would expect an increase of uptake of [18F]-AQ28A in visceral WAT vs SAT in DIO vs lean mice. It would be interesting to have these data included in the manuscript.

A19. The reviewer raises a valid point. However, due to the metabolism of [18F]-AQ28A by liver and storage in bladder, it is difficult to determine uptake in VAT and SAT (as these regions are very nearby confounding the analysis). Also, our MRI scans were performed with a limited field of view only accommodating the upper region of the animals. This was done to increase signal to noise and so a precise anatomical analysis of uptake in VAT and SAT is impossible. Finally, levels of Pde10a mRNA in these depots are significantly lower and our PET device may not have the sensitivity to measure such low levels of PDE10A.

Figure 4.

R20. Figure 4A-B. Why did the authors change the concentration of the inhibitor MP-10 from 30 to 10mg/kg? Moreover, how the authors explain that even vehicle-treated DIO mice are losing weight? Are these mice still in high fat diet?

A20. We chose the higher concentration for the metabolic cage studies to stimulate a robust effect on oxygen consumption as we only had a limited sample size which we could accommodate on this apparatus. Animals normally lose weight during chronic feeding studies in the initial stages due to the stress exposure, which was equal for both groups (Figure 4A and Supplementary Figure 9). However, from day 3, vehicle treated mice steadily regained weight whereas MP-10 mice continued to lose weight (Figure 4A and Supplementary Figure 9). During the feeding experiments, the mice continued to receive the same diet.

R21. Metabolic phenotyping is incomplete. Authors should investigate the effect of MP-10-induced weight loss/increased energy expenditure on insulin sensitivity.

A21. We performed ITTs to assess glucose homeostasis after chronic treatment with MP-10 in DIO mice. We indeed found that MP-10 treated mice were more insulin sensitive compared to vehicle treated controls (Figure 4C).

R22. Figure 4D. The Resting Energy Expenditure in this case should not be corrected by body weight. The mice treated with MP-10 are leaner with respect to the control mice, so forcedly the EE of these mice will result decreased vs untreated. To confirm that the decreased EE is not an artefact of the calculation, the EE should be normalised to the lean mass or assuming that the mice weight the same.

A22. We thank the reviewer for this comment and have amended this mistake and expressed energy expenditure without correcting for bodyweight (Figure 4D).

R23. Figure 4G. On top of thermogenic and beige markers the authors should measure in the SAT, VAT and BAT of the mice also the expression of PDE10A to validate the effect of MP-10. Once again, the authors should confirm VAT browning through histology analyses.

A23. We performed IHC analysis for UCPI in SAT, VAT and BAT samples after chronic treatment with MP-10 (Figure 4H). Our results are consistent with the gene expression analysis (Figure 4G).

Figure 5.

R24. The authors should provide also a representative PET image of before the human subject received the [18F]-MNI-1869 ligand as positive control. Moreover they should also confirm the specificity of this ligand [18F]-MNI-1869 for human PDE10A. The authors should add on the graph the [18F]-MNI-1869 uptake in the brain of human subjects. It would be also interesting to plot individual curves of [18F]-MNI-1869 uptake. Considering the fact that 2 of these patients are obese or overweight, it would be interesting to see if they display higher uptake of PDE10A radioligand.

A24. As described above, no baseline image is taken during PET scans. The specificity of [18F]-MNI-1869 was determined by the investigators in the previous publication (Barret et al., 2014 Journal of Nuclear Medicine). Uptake of [18F]-MNI-1869 by striatum was already performed in aforementioned study. However, we plotted individual curves for BAT and muscle and indeed found a clearly higher [18F]-MNI-1869 uptake by BAT in patients with higher BMI (Supplementary Figure 10).

Minor comments

R25. Figure 2. Why the authors perform the experiment both a fasted and ad libitum fed state? And not only a fed state?

A25. We have now explained our rationale for scans in the fed and fasted state in the methods section with reference to Fueger et al., 2006 Journal of Nuclear Medicine who showed differences between the two states (lines 507-512).

R26. In the second part of the paragraph entitled "PDE10A regulates brown and visceral white adipose tissue thermogenic function", the numbering of Figure 2 and Supplementary Figure 2 panels in the text with respect to the figures themselves is wrong. We ask the authors to check that carefully.

A26. We thank the reviewer for this comment and have carefully ensured that all references to figures in the text are correct.

Referee #2 (Remarks):

As stated by the authors, the current study describes an entirely novel role for PDE10A in adipose tissue and further support that selective PDE10A inhibitors may represent ideal treatment candidates for the management of obesity due to their dual ability to suppress feeding and increase energy expenditure.

Based the novelty of these data and the fact that they were able to show that expression and function of PDE10A in adipose tissue translated from mouse to humans, I believe this manuscript will have a high impact on the field.

R1. My only concerns stem from the lack of strong controls for the imaging studies. While the wealth of data argues that the biological effects they characterize in this study were driven by PDE10A, I believe the authors should have used PDE10A knockout mice to confirm their findings.

However, I feel the data in the manuscript still warrants publication without further studies. I suggest the authors implement my suggestions in future publication exploring this exciting new mechanism.

A1. We thank the reviewer for their very kind comments and for taking the time to review our manuscript. As already outlined in answers 5 and 8 to reviewer #1, our previous in vitro data to [18F]-AQ28A (Wagner et al., Eur J Med Chem, 2016) reveals highly selective binding of FAQ to PDE10a. We agree that KO mice are the gold standard method of confirming this and intend to use such mice in future studies. We did however perform a blocking experiment with MP-10 and did indeed find displacement of [18F]-AQ28A in both striatum and BAT (Supplementary Figure 2).

R2. A second concern is that the investigators used a very high dose of the PDE10A inhibitor MP-10 and it would be useful to run a dose titration of this compound to understand the PDE10A occupancy/ efficacy relationship in the model system.

A2. We agree that the dose used is high, but feel confident that other PDEs are not targeted due to the lack of changes in cyclic nucleotide and gene expression in SAT (Figure 2D).

Acceptance

25 April 2016

Please find enclosed the final reports on your manuscript. We are pleased to inform you that your manuscript is accepted for publication and is being sent to our publisher to be included in the next available issue of EMBO Molecular Medicine.

Congratulations on your interesting work,

***** Reviewer's comments *****

Referee #1 (Remarks):

We are satisfied the way the authors addressed our comments to the manuscript entitled "A novel thermoregulatory role for PDE10A in mouse and human adipocytes". We consider that the revised form of the paper is now suitable for publication in EMM.

Referee #2 (Comments on Novelty/Model System):

Please see previous review

Referee #2 (Remarks):

The authors adequately addressed my comments from the last review and the manuscript is now suitable for publication

Corresponding Author Name: Liliane Michalik

Manuscript Number: EMM-2015-05384